# Identification of miRNA–mRNA Regulatory Modules Involved in Lipid Metabolism and Seed Development in a Woody Oil Tree (*Camellia oleifera*)

**DOI:** 10.3390/cells11010071

**Published:** 2021-12-27

**Authors:** Bo Wu, Chengjiang Ruan, Asad Hussain Shah, Denghui Li, He Li, Jian Ding, Jingbin Li, Wei Du

**Affiliations:** 1Key Laboratory of Biotechnology and Bioresources Utilization, Ministry of Education, Institute of Plant Resources, Dalian Minzu University, Dalian 116600, China; wb512703926@163.com (B.W.); lihe@dlnu.edu.cn (H.L.); dingjian@dlnu.edu.cn (J.D.); lijingbin@dlnu.edu.cn (J.L.); duwei@dlnu.edu.cn (W.D.); 2Department of Biotechnology, Faculty of Sciences, University of Kotli Azad Jammu and Kashmir, Azad Jammu and Kashmir, Kotli 11100, Pakistan; syedasadhamdani@gmail.com; 3Guizhou Wulingshan Youcha Technology Innovation Research Institute Co., Ltd., Tongren 554300, China; lengfengtaolue@163.com

**Keywords:** *Camellia oleifera*, small RNA sequencing, miRNA–mRNA regulatory modules, lipid metabolism, seed development

## Abstract

Tea oil camellia (*Camellia oleifera*), an important woody oil tree, is a source of seed oil of high nutritional and medicinal value that is widely planted in southern China. However, there is no report on the identification of the miRNAs involved in lipid metabolism and seed development in the high- and low-oil cultivars of tea oil camellia. Thus, we explored the roles of miRNAs in the key periods of oil formation and accumulation in the seeds of tea oil camellia and identified miRNA–mRNA regulatory modules involved in lipid metabolism and seed development. Sixteen small RNA libraries for four development stages of seed oil biosynthesis in high- and low-oil cultivars were constructed. A total of 196 miRNAs, including 156 known miRNAs from 35 families, and 40 novel miRNAs were identified, and 55 significantly differentially expressed miRNAs were found, which included 34 upregulated miRNAs, and 21 downregulated miRNAs. An integrated analysis of the miRNA and mRNA transcriptome sequence data revealed that 10 miRNA–mRNA regulatory modules were related to lipid metabolism; for example, the regulatory modules of ath-miR858b–*MYB82/MYB3/MYB44* repressed seed oil biosynthesis, and a regulation module of csi-miR166e-5p–*S-ACP*-*DES6* was involved in the formation and accumulation of oleic acid. A total of 23 miRNA–mRNA regulatory modules were involved in the regulation of the seed size, such as the regulatory module of hpe-miR162a_L-2–*ARF19*, involved in early seed development. A total of 12 miRNA–mRNA regulatory modules regulating growth and development were identified, such as the regulatory modules of han-miR156a_L+1–*SPL4/SBP2*, promoting early seed development. The expression changes of six miRNAs and their target genes were validated using quantitative real-time PCR, and the targeting relationship of the cpa-miR393_R-1–*AFB2* regulatory module was verified by luciferase assays. These data provide important theoretical values and a scientific basis for the genetic improvement of new cultivars of tea oil camellia in the future.

## 1. Introduction

Tea oil camellia (*Camellia oleifera* Abel.) is an important woody oil tree in southern China and has a cultivation history of over two thousand years. Together with oil palm, olive, and coconut, tea oil camellia is well known as one of the four major woody oil trees in the world [1,2]. Tea oil camellia seeds have a high oil content of up to 40%; are rich in unsaturated fatty acids (the oleic acid content accounts for over 75% of all the fatty acids in the oil of the seeds); exhibit an abundance of bioactive compounds, such as vitamins, camelliaside, and tea polyphenols; and are an important source of high-grade edible vegetable oil [3]. The oil extracted from tea oil camellia seeds is not easily oxidized, as it has high chemical stability and can be preserved for a long time; it can significantly contribute to the prevention of cardiovascular diseases and cancer and reduce cholesterol and blood lipids [4,5]. However, the seed yield of tea oil camellia is not stable, and the seed oil content varies greatly among different germplasms, which seriously restricts the sustainable development of the tea oil camellia industry. One important strategy is the breeding of tea oil camellia cultivars with high seed oil contents and high and stable seed yields.

MicroRNAs (miRNAs) are a class of endogenous noncoding small RNAs with lengths of approximately 20–25 nucleotides, and they play a crucial role in mediating the posttranscriptional regulation of gene expression by targeting mRNAs for degradation or inhibiting their translation in eukaryotes. MiRNAs are derived from primary miRNA transcripts (pri-miRNAs) that contain a stem–loop secondary structure; the pri-miRNA is processed in the nucleus by DCL1, a Dicer-like protein, to create an miRNA–miRNA* duplex, where miRNA* is a passenger strand complementary to the miRNA. The duplex is then separated by helicase, and the mature miRNA is incorporated into an ARGONAUTE 1 (AGO1) protein to form an RNA-induced silencing complex (RISC) [6,7]. Target genes that contain a sequence with almost complete complementarity to the miRNA are cleaved by the RISC at a specific site opposite to the 10th or 11th nucleotide in the miRNA [8]. Studies have confirmed the key roles of miRNAs in various biological and metabolic processes in plants, such as fatty acid biosynthesis [9,10], lipid metabolism [11,12], growth and development [13], responses to stresses [14], and cellular proliferation and differentiation [15]. As of 3 December 2021, a total of 38,589 miRNA sequences have been deposited in the database, miRBase [16,17], and many conserved and novel miRNAs have been identified in soybeans [18], *Arabidopsis* [19], *Brassica napus* [20], rice [21], peonies [22], and maize [23]. In walnuts, jre-miRn105, jre-miRn434, jre-miR477d, and jre-miR156a.2 are key miRNAs that regulate FA synthesis, and jre-miRn411 and jre-miR399a.1 are closely related to oil accumulation [10]. MiRNAs control and influence a variety of physiological processes by regulating target genes. For example, the *KAS* and *KAR* genes, targeted by miR159 and miR156, respectively, are important for lipid biosynthesis [24]. The *FAD*2 gene, targeted by miR159b and miR5026, regulates and influences FA biosynthesis [18,25]. Enhancing microRNA167A expression in the seeds decreased the alpha-linolenic-acid content and increased the seed size in *Camelina sativa* [26]. MiR172 plays a highly important role in seed development by modulating its target *ARF2* gene [11].

Transcription factors play critical roles in regulating lipid biosynthesis [27], such as WRINKLED1 and LEAFY COTYLEDON, MYB, SPL, ARF, and AP2 [28], which have been identified as targets of the major miRNAs [29]. On the basis of the high-throughput small RNA and degradome analyses of soybean seeds 15 days after flowering, 55 annotated and 26 novel miRNAs, which targeted 145 genes, were identified [18]; a total of 82% of the targeted genes were transcription factors, including the ARF, MYB, TCP, GRF, and NAC families [18]. Known transcription-factor-encoding genes involved in seed size/weight determination, including *SPL*, *GRF*, *MYB*, *ARF*, *HAIKU*1, *SHB*1, *KLUH*/*CYP78A5*, and *E2Fb*, along with novel genes, were found to be targeted by the predicted miRNAs in chickpeas [30]. MiRNA–target expression profiles evidenced that some miRNAs could tune distinct seed developmental stages by targeting the HD-ZIP, ARF, SPL, and NF-Y transcription-factor families in *Phaseolus vulgaris* [31]. MiR160 negatively regulates ARFs that significantly affect seed development in *A. thaliana* [32]. SPL10 and SPL11 are targeted by miR156 and are involved in the early morphogenesis of *Arabidopsis* embryos [33]. MYBs targeted by miR159 are involved in seed size, and a double mutation of miR159 (miR159ab) increased MYB33 and MYB65 expression, which promoted the formation of small seeds [34]. The transcription factor, WRI1, targeted by hrh-miRn215 in sea buckthorn seeds, involved in oil biosynthesis and fatty acid formation and accumulation [35], ethylene-responsive transcription factor 3 (ERF3), MADS domain protein, AGAMOUS-LIKE 61 (AGL61), and WRKY transcription factor 41 (WRKY41) were targeted by miR171k-5p_1, miR7760-p3_1, and Xso-miRn80 in the seeds of *Xanthoceras sorbifolium* [11].

Currently, miRNA–mRNA regulatory modules controlling multiple biological and metabolic processes have been revealed via the functional analysis of woody oil plants, such as hazel [36], oil palm [37], olive [38], peony [22], walnut [10], and yellowhorn [11]. In developing sea buckthorn seeds, 19 (4 known and 15 novel) and 22 (6 known and 16 novel) miRNAs were found to be involved in lipid biosynthesis and seed size, respectively [39], and an integrated analysis of the mRNA and miRNA transcriptomes and the qRT-PCR revealed key miRNAs and their targets (miR164d-*ARF2*, novelmiRNA-108-*ACC*, novelmiRNA-23-*GPD*1, novelmiRNA-58-*DGAT*1, and novelmiRNA-191-*DGAT*2) that are potentially involved in the seed size and lipid biosynthesis in the seeds of sea buckthorn. In developing seeds of yellowhorn, 141 differentially expressed miRNAs (120 known and 21 novel miRNAs) were identified, along with several miRNA–target regulatory modules involved in the lipid biosynthesis of yellowhorn seeds, such as miR319p_1-*FAD*2–2, miR5647-p3_1-*DGAT*1, and miR7760-p5_1-*MED*15A [11]. However, there are no reports on the roles of miRNAs in the different developmental stages of the seeds of high- and low-oil cultivars of tea oil camellia. 

To understand the potential roles of miRNAs and their target genes in the developing seeds of tea oil camellia, high-throughput Illumina sequencing technology was used to reveal the global profile of small RNAs in the developing seeds of high- and low-oil tea oil camellia cultivars. Then, we identified some novel and known miRNAs and their target genes by the integrated analysis of the mRNA and miRNA transcriptomes. Finally, we further analyzed differentially expressed miRNAs and the GO and KEGG enrichments of these miRNAs target genes. In addition, reverse transcription–quantitative PCR (qRT-qPCR) was used to validate the sequencing results for a selected group of miRNAs and the expression profiles of their targets, and the targeting relationship of the miRNA–mRNA regulatory module was verified by luciferase assays. Our datasets and results will promote the development of tea oil camellia miRNAs in public resource databases, and will provide a scientific resource and basis for further investigations of the identified miRNAs of tea oil camellia and their regulatory roles in the genetic improvement of high-quality and high-yielding tea oil camellia in the future.

## 2. Materials and Methods

### 2.1. Study Site and Sampling

The sample trees were from two cultivars of tea oil camellia, the cultivar, ‘M3’, with a high seed oil content, and the cultivar, ‘M8’, with a low seed oil content (voucher No. 1303 and No. 1308, respectively, identified by Chengjiang Ruan and deposited at Dalian Minzu University) growing in the tea oil camellia orchard in Yuping Dong Autonomics County, Guizhou Province, China. It is located at 27°28′–27°31′ north latitude, and 108°34′–109°09′ east longitude. The orchard had a mean annual rainfall of 1174.1 mm, a mean annual temperature of 16.4 °C, and a mean annual sunshine rate of 1227.8. These two cultivars were selected from the tea oil camellia orchard because they had very similar phenotypic characteristics and genetic backgrounds, but not seed oil content.

For tea oil camellia, the key period of seed oil biosynthesis and accumulation is from June to September [40]. The oil contents in the seeds of these two cultivars showed a gradually upward trend from early June to early July; then, they began to rise rapidly from early July to early September and, subsequently, remained stable. The fruits were collected at random from the different plants of both cultivars on 2 June, 4 July, 5 August, and 3 September 2016, which were the same dates as those for the published mRNA-seq data [41], and the four sampling periods were referred to as T1~T4, for each cultivar in the data analysis; thus, we recorded them as M3T1, M3T2, M3T3, and M3T4, and as M8T1, M8T2, M8T3, and M8T4 for the samples of the two cultivars of the four periods, respectively. The seeds obtained by stripping the fruit shells were immediately wrapped in tin foil and placed into liquid nitrogen. The frozen seeds were transported to the laboratory and stored at −80 °C for the following experiments. The collection of all the samples fully complied with local and national legislation. 

### 2.2. Small RNA Library Construction and Sequencing

The total RNA of each sample was extracted by using TRIzol reagent (Invitrogen, Carlsbad, CA, USA), according to the manufacturer’s instructions. The quality and quantity of the total RNA of each sample were analyzed using the Bioanalyzer 2100 (Agilent, Santa Clara, CA, USA) and RNA 6000 Nano LabChip Kit (Agilent, Santa Clara, CA, USA), ensuring that the RIN (RNA integrity number value) was >7.0. The small RNA library for each sample was constructed with the TruSeq Small RNA Sample Prep Kit (Illumina, San Diego, CA, USA), in accordance with the manufacturer’s protocol. Then, we performed single-end sequencing (36 bp or 50 bp) on an Illumina HiSeq2500 at LC-BIO (Hangzhou, China), following the vendor’s recommended protocol.

### 2.3. Identification and Analysis of miRNA Sequencing Data

To obtain accurate and reliable miRNA sequencing data, the raw read quality was evaluated using Illumina FastQC to obtain Q30 data. Then, the raw reads were processed to obtain valid reads with an in-house program, ACGT101-miR (LC Sciences, Houston, TX, USA) [42], to filter out adapter-dimer reads, junk reads, low-quality reads, low-complexity reads, reads for common RNA families (rRNA, tRNA, snRNA, and snoRNA), and repeats. Then, the unique sequences, with lengths of 18–25 nt, were selected to map to the precursors of 66 specific species (Appendix A) in miRBase 21.0 (http://www.mirbase.org/, accessed on 3 August 2017)), through a BLAST search, to identify known miRNAs and novel 3p- and 5p-derived miRNAs [43]. Because miRNAs are conserved, related species of tea oil camellia were selected as specific species. In the alignment analysis, the seed length was set to 16, and the length variation at both the 3′ and 5′ ends, and one mismatch inside the sequence, were allowed. The unique sequences mapping to the hairpin arms of the mature miRNAs of specific species were identified as known miRNAs, and the unique sequences mapping to the other arms of known specific species’ precursor hairpins opposite to the annotated mature miRNA-containing arms were considered to be novel 5p- or 3p-derived miRNA candidates [44]. The remaining sequences were mapped to other selected species precursors (with the exclusion of specific species) in miRBase 21.0 by a BLAST search, and the mapped pre-miRNAs were further BLAST-searched against the specific species genomes to determine their genomic locations. We defined the above two as known miRNAs. The unmapped sequences were BLAST-searched against specific genomes, and the hairpin RNA structures containing sequences were predicted from the flanking 120 nt sequences using the RNAfold software (http://rna.tbi.univie.ac.at/cgi-bin/RNAfold.cgi, accessed on 13 October 2017)). The criteria for the secondary structure prediction were: (1) The number of nucleotides in one bulge in the stem (≤12); (2) The number of base pairs in the stem region of the predicted hairpin (≥16); (3) The cutoff of free energy (kCal/mol ≤−15); (4) The length of the hairpin (up and down stems + terminal loop ≥50); (5) The length of the hairpin loop (≤200); (6) The number of nucleotides in one bulge in the mature region (≤4); (7) The number of bias errors in one bulge in the mature region (≤2); (8) The number of biased bulges in the mature region (≤2); (9) The number of errors in the mature region (≤4); (10) The number of base pairs in the mature region of the predicted hairpin (≥12); and (11) The percent of mature sequence in the stem (≥80). After alignment analysis, the miRNAs that met the above criteria were considered novel miRNAs. 

### 2.4. Analysis of Differentially Expressed miRNAs

The number of clean reads originating from each miRNA represents the relative expression abundance, or the level of the corresponding miRNA in small RNA deep sequencing. Normalized read counts were calculated for miRNA differential expression analysis according to the procedure in a previous study [45], with minor modifications: (1) Find a set of sequences common among all the samples; (2) Construct a reference dataset. Each datum in the reference set is the copy number median value of a corresponding common sequence of all the samples; (3) Perform base-2 logarithm transformation on the copy numbers (*log*_2_(*copy#*)) of all the samples and reference the dataset; (4) Calculate the (*log*_2_(*copy#*)) difference (∆*log*_2_(*copy#*)) between the individual sample and the reference dataset; (5) Form a subset of sequences by selecting *|*∆*log*_2_(*copy#*)| < 2, which means less than a (2^2^ =) four-fold change from the reference set; (6) Perform linear regressions between individual samples and the reference set on the subset sequences to derive linear equations, y = *a*_i_x + *b*_i_, where *a*_i_ and *b*_i_ are the slope and intercept, respectively, of the derived line; x is the *log*_2_(*copy#*) of the reference set; and y is the expected *log*_2_(*copy#*) of sample i on a corresponding sequence; (7) Calculate the mid value, x_mid_ = (max (x) − min (x))/2 of the reference set. Calculate the corresponding expected *log*_2_(*copy#*) of sample i, y_i,mid_ = *a*_i_x_mid_ + *b*_i_. Let y_r,mid_ = x_mid_, and let ∆y_i_ = y_r,mid_ − y_i,mid_, which is the logarithmic correction factor for sample i. We then derive the arithmetic correction factor, *f*i = 2^Δyi^, for sample i; (8) Correct the copy numbers for the individual samples by multiplying the corresponding arithmetic correction factor, *f*i, by the original copy numbers. 

A Student’s *t*-test was used to identify miRNAs differentially expressed between two samples on the basis of the normalized read counts. An analysis of variance (ANOVA) was used to identify miRNAs differentially expressed among samples at the four development stages in each line by using the mirnaTA software [46]. The miRNAs with *p*-values < 0.05 were considered differentially expressed miRNAs [45]. The normalized read counts of some miRNAs were set to be 0.01 for further calculation if they had no reads in the library.

### 2.5. Prediction and Identification of miRNA–mRNA Regulatory Modules

The target genes of differentially expressed miRNAs were identified by aligning mature miRNA sequences with published mRNA-seq sequences [41] using TargetFinder (https://github.com/carringtonlab/TargetFinder, accessed on 8 April 2018) [47], following the procedures described in previous studies [48]. The TargetFinder algorithm is based on the miRNA and mRNA complementary-pairing principle. The alignment score value in the prediction result was used as the screening standard, which was the score of the prediction result. The comparison between the miRNA and mRNA was performed, and then the complementary pairing score was given for the comparison position: a total of 1 point was given for one mismatch or deletion, and 0.5 points were given for one G:U pairing; the penalty for a mismatched G:U pairing in the core region was doubled, and the core region began at the 2nd nt and continued to the 13th nt of the miRNA (5′–3′). Finally, the results were obtained for scores less than or equal to 4. The significance of the alignment score was the degree of match between the target gene and the miRNA. The lower the score, the more complete the matching, and the more credible it was. 

The GO terms and KEGG pathways of genes targeted by these differentially expressed miRNAs were annotated. The GO function and KEGG information of this species were used to perform GO function annotation (http://bioinfo.cau.edu.cn/agriGO/, accessed on 1 September 2018) and KEGG signaling pathway annotation (https://www.kegg.jp/kegg/pathway.html, accessed on 1 September 2018) for the target genes of differentially expressed miRNAs. Fisher’s exact hypothesis test was performed on the GO and KEGG pathways of the targeted genes, and the enrichment analysis of each GO term and KEGG pathway was separately performed. According to the prediction results for the targeted genes of the differentially expressed miRNAs, significance differences in the GO and KEGG information were defined with a default threshold, *p* < 0.05.

### 2.6. Identification of the Expression of miRNAs and Their Target Genes by qRT-PCR

To validate the high-throughput sequencing data, six pairs of miRNA target genes, on the basis of the sequencing data, were selected to perform the qRT-PCR. The total RNA from each sample for the qRT-PCR analysis was extracted, as described above, for the sequencing experiments. First-strand cDNA was synthesized using the Mir-X miRNAs First-Strand Synthesis Kit (TaKaRa Biotech, Dalian, China), according to the manufacturer’s instructions. The U6 snRNA was selected as the reference gene, and the qRT-PCR for the miRNA experiments was performed in an ABI7500 real-time PCR instrument (Applied Biosystems, Foster City, CA, USA), using the Mir-X miRNA qRT-PCR SYBR Kit (TaKaRa Biotech, Dalian, China) and miRNA-specific primers (Appendix A). The qRT-PCR experiments for each sample were performed with three replicates.

The predicted target genes of the miRNAs were selected for qRT-PCR analyses. First-strand cDNA was synthesized using the PrimeScript^TM^ RT Master Mix (TaKaRa Biotech, Dalian, China), according to the manufacturer’s instructions. All the specific primer pairs of the predicted target genes (Appendix A) were designed using the Primer Quest online software (http://sg.idtdna.com/PrimerQuest/Home/Index, accessed on 10 September 2018). Elongation factor 1-alpha (*CoEF1*α) was selected as a reference gene. The qRT-PCR experiments were performed using the SYBR Premix Ex Taq^TM^ II Kit (Tli RNaseH Plus; TaKaRa Biotech, Dalian, China). The qRT-PCR was conducted using the ABI7500 real-time PCR instrument (Applied Biosystems, Foster, CA, USA). The qRT-PCR analyses for each sample were performed in triplicate. The relative expression levels of each gene were calculated using the 2^−ΔΔCt^ method [49].

### 2.7. Dual-Luciferase Reporter Assay

Many studies have shown the AFB2 gene and miR393 to be involved in plant growth and development [50,51], but few have reported on the potential role of the miR393–AFB2 regulatory module in seeds. Thus, we investigated the targeting relationship between cpa-miR393_R-1 and AFB2 in the seeds of tea oil camellia. Fragments from the 3′-UTR of AFB2 containing the predicted binding sequences for cpa-miR393_R-1 were amplified and subcloned into the pmirGLO luciferase promoter vector. The pCDNA3.1 plasmid was used as the template vector. The fragment containing the nucleotide sequences of the precursor of cpa-miR393_R-1 was cloned into the vector to construct the recombinant vector expressing cpa-miR393_R-1 pCDNA3.1, as described previously. The pmirGLO vector containing the 3′-UTR of AFB2 was cotransfected with pCDNA3.1 or pCDNA3.1, containing pre-cpa-miR393_R-1 into HEK-293T cells [52], using Lipofectamine 2000 (Invitrogen, Carlsbad, CA, USA), according to the manufacturer’s protocol and a previous report [53,54,55]. Forty-eight hours after treatment, the firefly and Renilla luciferase activities were measured using a luciferase reporter assay kit (BioVision, Inc., Milpitas, CA, USA). Renilla was used as the transfection control.

## 3. Results

### 3.1. Brief Outline for Sequencing Data, from Raw Data to Cleaned Sequences

To comprehensively identify key small RNAs in the developing seeds of high- and low-oil cultivars of tea oil camellia, 16 small RNA libraries were sequenced on an Illumina HiSeq2500 platform for four different seed developmental stages, with two biological replicates for each seed developmental stage of each cultivar. To evaluate the consistency of the biological replicates, the Pearson correlation between the samples was calculated and is shown in Appendix A. A high correlation coefficient (*r*) between every two replicates confirmed the reliability, as well as the operational stability, of the experimental results. Totals of 13,572,633 and 14,183,198 raw reads were obtained from the cultivars, ‘M3’ and ‘M8’, of tea oil camellia, respectively. Furthermore, the raw reads were analyzed with the LC sRNA analysis pipeline, ACGT101-miR, and totals of 11,653,829 (85.67%) and 12,248,627 (86.23%) valid reads were obtained for ‘M3’ and ‘M8’ after removing the adaptor dimers, junk reads, low-complexity sequences, and sequences shorter than 18 nucleotides, or longer than 25 nucleotides, respectively (Appendix A). The length distributions of the unique sRNAs for both cultivars at four developmental stages were then summarized. Most of the sRNA reads ranged from 21 to 24 nt in length, leading to similar length distributions in two tea oil camellia cultivars at the different developmental stages. sRNAs of 24 nt were the most abundant, accounting for 71.58% (M8T1) to 79.74% (M3T4) of the total (Figure 1). In addition, 21-, 22-, 23-, and 25-nt sRNAs were common, which were more abundant than those of any other length, besides 24 nt. 

### 3.2. Identification of Known and Novel miRNAs in Developing Seeds of Tea Oil Camelia

To detect the miRNAs involved in the development of tea oil camellia seeds, the unique sequences were selected and mapped against miRBase 21.0 by a BLAST search to identify the known miRNAs and novel miRNAs, and then a total of 196 expressed miRNAs were identified in the developing seeds of tea oil camellia (Appendix A). The expressed miRNAs were classified into five groups (gp1, gp2a, gp2b, gp3, and gp4), according to criteria defined in a previous study [44], with minor modifications: seven known miRNAs belonged to gp1; 149 conserved and known miRNAs belonged to gp2a, gp2b, and gp3; and 40 novel miRNAs belonged to gp4 (Appendix A). Analyses of the number and length ratios of unique miRNA sequences of different lengths revealed that the lengths of the miRNAs were mainly distributed within 20~24 nt (Appendix A). MiRNAs with a length of 21 nt were frequent, and accounted for 49.55% of all the miRNAs, which is consistent with the definition of miRNAs (Appendix A). The results of the statistical analysis of the expression levels of the detected miRNAs, and the detection rates for miRNAs upon evaluating redundant miRNAs, showed that 196 unique miRNAs were identified, including 156 known miRNAs (Appendix A), and 40 novel miRNAs (Appendix A). 

The miRNA sequences were mapped to known miRNAs from 42 plant species, with the highest number of miRNAs mapped to known gma-miRNAs of *Glycine max* (17 (10.89%)), followed by ath-miRNAs of *Arabidopsis thaliana* (13 (8.33%)), and aly-miRNAs of *Arabidopsis lyrata* (11 (7.05%)) (Appendix A). Among the identified known miRNA sequences, 156 were identified as belonging to 35 miRNA families, and the MIR171 family had the largest number of members (12), followed by the MIR167 (11), and the MIR156 and MIR5645 families, with 10 members each (Appendix A). The analysis of the distribution of the miRNA first-nucleotide preferences showed that miRNAs of 24 nt tended to begin with 5′-A in tea oil camellia, while 21–22-nt miRNAs tended to start with 5′-U, and 23-nt and 25-nt miRNAs tended to start with 5′-G (Figure 2a). At the 5′ end, uridine (26.12%) and adenine (26.06%) were the similar common nucleotides for all the known miRNAs in tea oil camellia (Figure 2b). The expression profiles of the known miRNAs showed that the normalized read counts of the known miRNAs varied from 118,375 to less than 10 reads, exhibiting great variation, even within the same family. The normalized expression data were sorted from low to high. If the copy number for an identified miRNA was less than 10 in all the samples, or higher than 10 in one sample and less than the average copy number of the dataset in all the samples, this indicated that it had low expression; if the copy number of an identified miRNA was higher than the average copy number of the dataset in any sample, this indicated that it was highly expressed. According to the above classification method, 38 known miRNAs were highly expressed, and all the novel miRNAs had low expression. A total of 4 of the 40 miRNAs with low expression had copy numbers less than 10 in all the samples: col-miRn-5p-236956_11; col-miRn-3p-145806_25; col-miRn-3p-253727_10; and col-miRn-3p-150080_24. Compared to the known miRNAs, most of the identified novel miRNAs had relatively low read counts (normalized), and only five novel miRNAs had more than 100 reads in all the libraries: col-miRn-5p-1103_1978; col-miRn-5p-2142_1228; col-miRn-3p-18781_243; col-miRn-3p-3803_812; and col-miRn-3p-18137_250. The secondary structures of these five most abundant novel miRNAs were predicted (Appendix A), indicating that they can form typical stem–loop hairpins, and that their folding free energies were −62.4, −71.9, −71.9, −50.0, and −66.9 kcal·mol^−1^, respectively. 

In addition, we detected the number of specifically expressed miRNAs in one sample, and the number of specifically expressed miRNAs simultaneously detected in two or more samples, which were visualized using a Venn diagram, comparing the samples of the two cultivars of ‘M3’and ‘M8’ at four different developmental stages (Figure 3). It was found that 54 miRNAs were identified at four different seed developmental stages of the ‘M3’ cultivar (Figure 3a), and 63 miRNAs were identified for the ‘M8’ cultivar (Figure 3b). Mining these simultaneously expressed miRNAs could provide a favorable foundation for the genetic improvement of tea oil camellia.

### 3.3. Identification of Differentially Expressed miRNAs

The differentially expressed miRNAs were identified using suitable differential test methods, in which significantly differentially expressed miRNAs were defined with a significance threshold of a *p*-value < 0.05. There were 55 significantly differentially expressed miRNAs identified between the samples of each cultivar at two different developmental stages, and between different samples of two cultivars at the same developmental stage. A total of 34 upregulated miRNAs, and 21 downregulated miRNAs, were identified among these significantly differentially expressed miRNAs (Appendix A). Moreover, to identify the expression level differences of the miRNAs significantly differentially expressed between the high- and low-oil cultivars of tea oil camelia, the significantly differentially expressed miRNAs were compared between the seeds of the cultivars of ‘M3’ and ‘M8’ at the same development stage (Figure 4, Appendix A). At the early seed developmental stage, five miRNAs (mtr-MIR2586a-p3_1ss15TC; ath-MIR5645b-p5_2ss19TG21TC; ath-MIR5645b-p3_2ss19TG21TC; mtr-MIR2586a-p3_2ss4TG21CT; and hpe-miR166a_1ss5AC) were upregulated in ‘M3’ relative to ‘M8’, and hpe-miR166a_1ss5AC had a 0.94-fold change (log2(ratio)). At the second seed developmental stage, three miRNAs (csi-miR1515_1ss16AG; stu-miR8016_1ss7GT; and ppe-miR172a-5p_1ss21GA) were upregulated in ‘M3’ relative to ‘M8’, with 0.38-, 1.86-, and 2.57-fold changes (log2(ratio)), respectively. Seven miRNAs (rco-miR156e_L+1R-1; dpr-miR167a_R+1; ccl-miR167a,tcc-miR390a; ath-miR390a-5p; aly-miR167b-3p_1ss13AG; and han-miR156a_L+1) were downregulated, with -3.40-, -1.88-, -1.66-, -1.40-, -1.40-, -1.35-, and -2.81-fold changes (log2(ratio)), respectively. At the late seed developmental stage, four miRNAs (col-miRn-3p-9080_426; mtr-MIR2586a-p3_1ss15TC; ghr-MIR7499-p3_2ss10GC17CG; and rco-miR167a_R+1, with -0.33-fold changes (log2(ratio)), were downregulated in ‘M3’ compared with their expression in ‘M8’.

In addition, the number of common and specific differentially expressed miRNAs is visually shown in a Venn diagram comparison between different comparison groups (Figure 5). The results show that there were four significantly differentially expressed miRNAs (csi-miR1515_1ss16AG; ccl-miR167a; aly-miR167b-3p_1ss13AG; and ppe-miR172a-5p_1ss21GA) simultaneously found in different comparison groups, when the significantly differentially miRNAs among different development stages in the ‘M3’ seeds were compared with those of the comparison of M3T2 vs. M8T2 (Figure 5a). There were six significantly differentially expressed miRNAs (csi-miR1515_1ss16AG; ccl-miR167a; aly-miR167b-3p_1ss13AG; ppe-miR172a-5p_1ss21GA; aly-miR167b-3p_1ss13AG; and ppe-miR172a-5p_1ss21GA) simultaneously found in different comparison groups, when the significantly differentially miRNAs among different development stages in the ‘M8’ seeds were compared with those of the comparison of M3T2 vs. M8T2 (Figure 5b). Moreover, six significantly differentially expressed miRNAs (aly-miR167b-3p_1ss13AG; ptc-miR172b-5p; ssl-miR828; csi-miR1515_1ss16AG; ath-miR858b; and ppe-miR172a-5p_1ss21GA) were simultaneously found in the comparison of the whole ‘M3’ and ‘M8’ seed developmental groups (Figure 5c), but only one differentially expressed miRNA (rco-miR167a_R+1) was simultaneously found between the comparison of the whole ‘M3’ seed developmental group and the comparison of M3T4 vs. M8T4 (Figure 5d). Mining for these significantly differentially expressed and simultaneous miRNAs is helpful for exploring the expression differences and regulatory patterns between the seeds of high- and low-oil cultivars of tea oil camellia at different developmental stages, which could provide a scientific basis for improving the seed oil content and quality of tea oil camellia in the future. These results suggest that miRNA-mediated regulatory mechanisms may play an important role in various biological pathways during seed development in tea oil camellia.

### 3.4. Prediction and Identification of miRNA–mRNA Regulatory Modules

To further understand the functions of miRNAs in the developing seeds of high- and low-oil cultivars of tea oil camellia, the target genes of the differentially expressed miRNAs were predicted using the TargetFinder software on the basis of the miRNA sequences with published mRNA transcriptome data [41]. A total of 17,166 target genes were predicted, and some genes were targeted by multiple miRNAs, resulting in a total of 33,418 predicted miRNA–mRNA regulatory modules (Appendix A). Furthermore, the predicted target genes of known and novel miRNAs were subjected to gene ontology (GO) analysis. The results show that the target genes were classified into three categories by the hypergeometric distribution algorithm (Figure 6a): biological process, cellular component, and molecular function. A total of 25,988 miRNA–mRNA regulatory modules were related to various biological processes, 21,132 of which were found to be involved in different cellular components, and 14,675 of which were identified to be involved in molecular function (Figure 6a). The major biological functions of differentially expressed target genes were identified by the significant terms in the GO enrichment analysis; these results show that a total of 107 significant GO terms were related to the nucleus, sequence-specific DNA binding, and the regulation of transcription, the DNA-template, and electron carrier activity (Figure 6b and Appendix A). 

To further understand the biological functions of the target genes in the developing seeds of tea oil camellia, the target Kyoto Encyclopedia of Genes and Genomes (KEGG) pathway genes were identified by KEGG pathway enrichment analysis. These results show that a total of 16,871 miRNA–mRNA regulatory modules were involved in 206 individual KEGG pathways, involved in linoleic acid metabolism, alpha-linolenic acid metabolism, and glycosylated biosynthesis. The significant KEGG pathway enrichment analysis showed that 31 significant KEGG pathways (Figure 7 and Appendix A) were mainly involved in pyruvate metabolism, the regulation of the actin cytoskeleton, fatty acid biosynthesis, and the citrate cycle.

### 3.5. MiRNA–mRNA Regulation Modules Involved in Lipid Metabolism

The analyses of the differentially expressed miRNAs and their target genes involved in lipid metabolism revealed that multiple KEGG pathways were related to lipid metabolism during the seed development of tea oil camellia, according to the KEGG enrichment analysis (*p* < 0.05), in which 1869 miRNA–mRNA regulatory modules were involved in 12 pathways related to lipid metabolism, including 1723 predicted known miRNA–mRNA regulatory modules, and 146 novel miRNA–mRNA regulatory modules. In these regulation modules, the same miRNA often targeted different genes, and several miRNAs also targeted the same gene. There were 188 target genes involved in the fatty acid biosynthesis pathway, 411 target genes involved in the glycolysis/gluconeogenesis pathway, 118 target genes involved in the biosynthesis of the unsaturated fatty acid pathway, and 159 targeted genes related to the fatty acid metabolism pathway (Appendix A). The significant KEGG enrichment analysis and function annotation were integrated, and five significant KEGG pathways related to lipid metabolism were identified, including pyruvate metabolism, fatty acid biosynthesis, glycolysis/gluconeogenesis, sphingolipid metabolism, and steroid biosynthesis. 

A total of ten key miRNA–mRNA regulatory modules related to the lipid metabolism pathway were identified, including one novel miRNA (col-miRn-5p-21064_221), and eight known miRNAs that were involved in developing seeds of high- and low-oil cultivars of tea oil camellia (Table 1). These target genes were commonly involved in the seed oil biosynthesis pathway (Appendix A). In the fatty acid biosynthesis metabolic pathway (ko00061), the *ACC1* (comp59939_c1) gene was targeted by tcc-miR162. In the pyruvate metabolism metabolic pathway (ko00620), the *KAS1* (comp67779_c0) and *S-ACP*-*DES6* (comp67006_c0) genes were targeted by csi-miR166e-5p; the *KAS3B* (comp61049_c0) gene was targeted by byrgl-miR5139_L-1; the *fabG* (comp63911_c0) gene was targeted by mtr-miR156h-3p_1ss8AC; the *Mcat* (comp63026_c1) gene was targeted by cpa-miR164d; the *FATB1* (comp67050_c0) gene was targeted by col-miRn-5p-21064_221; the *MOD1* (comp52017_c0) gene was targeted by aly-miR393a-3p_1ss12TC; and the *SAD* (comp48800_c0) gene was targeted by ath-miR172a. The gma-MIR5368-p3_1ss21CT targeted the *GAPN* (comp67185_c0) gene, involved in the glycolysis/gluconeogenesis metabolic pathway (ko00010). 

The expression patterns and expression differences of the above identified miRNAs and their target genes related to lipid metabolism were determined by the heat map analysis of their expression levels in the developing seeds of high- and low-oil cultivars of tea oil camellia (Figure 8). The heat map of the miRNAs and their target genes was produced using the MEV software, ver 4.9.0., on the basis of the original expression levels of the miRNAs, and their target genes were normalized by z-score normalization. On the one hand, at the developing seed levels of the ‘M3’ cultivar, the expression of cpa-miR164d was the highest at the early seed development stage, and it then showed a declining trend; however, the expression of its target gene, *Mcat* (comp63026_c1), was the lowest at this stage, and it then gradually increased. These results indicate that there may be a negative correlation between cpa-miR164d and the expression of its target gene, *Mcat*. The expression of mtr-miR156h-3p_1ss8AC was significantly downregulated in this stage, and then gradually increased; however, its target gene, *fabG* (comp63911_c0), was the highest, and it then gradually decreased to the lowest expression level in the late developmental stage. In addition, from the second to third seed developmental stages of the ‘M3’ cultivar, the expression of four miRNAs (aly-miR393a-3p_1ss12TC; rgl-miR5139_L-1; col-miRn-5p-21064_221; and ath-miR172a) decreased gradually, while the expression of their target genes, *Mcat*, *KAS3B*, *FATB1,* and *SAD*, gradually increased.

On the other hand, the comparative analysis of the expression levels of these miRNAs and their target genes between the two cultivars showed that the expression of cpa-miR164d was significantly higher in ‘M3’ than in ‘M8’, but that that of mtr-miR156h-3p_1ss8AC was significantly lower in ‘M3’ than in ‘M8’ at the early seed developmental stage. The expression patterns of csi-miR166e-5p and Gma-MIR5368-p3_1ss21CT were similar in the developing seeds of the two cultivars of tea oil camellia. The former was similarly upregulated in the two cultivars, and its target genes, *KAS1* (comp67779_c0) and *S-ACP*-*DES6* (comp67006_c0), also showed a trend of upregulation. The latter showed low expression at the early seed developmental stage of the two cultivars, and its target gene, *GAPN* (comp67185_c0), also appeared to have a similar expression pattern in these two cultivars, with downregulated levels at the early seed developmental stage, and upregulated levels at the late seed developmental stage.

On the basis of these results, it could be speculated that these miRNAs and their target genes might be involved in the lipid metabolism pathways. The analysis of their interactions and influence in different relevant lipid metabolism pathways in tea oil camellia will provide important information for the breeding of tea oil camellia in the future.

### 3.6. MiRNA–mRNA Regulation Modules Involved in Seed Size

Seed size is one of the main factors affecting crop yield. A total of 23 miRNA–mRNA regulatory modules involved in seed size were predicted (Table 2 and Figure 9). Their target genes were transcription factors belonging to the MYB, CNR, MED, and ARF families, which play important roles in regulating seed development (Appendix A). On the one hand, Mdm-miR167h_1ss22AT showed a similar expression pattern in the ‘M3’ and ‘M8’ cultivars, with low expression levels in the early seed developmental stages, and high levels in the late seed developmental stages. Its target gene, *MED27* (comp135267_c0), showed the highest expression in the early seed developmental stage, and the lowest in the late seed developmental stage in the ‘M3’ cultivar; however, the expression of *MED27* remained high at all the seed developmental stages in the ‘M8’ cultivar. The expression of mtr-miR156h-3p_1ss8AC was the lowest at the early seed developmental stage in the ‘M3’ and ‘M8’ cultivars, and its target gene, *MYB44* (comp55270_c0), showed the highest expression in this stage. 

On the other hand, the expression patterns of most of the miRNAs were slightly different between the two cultivars, ‘M3’ and ‘M8’, such as col-miRn-5p-315953_8, ath-miR858b, and hpe-miR162a_L-2. For col-miRn-5p-315953_8, its expression was relatively low in four seed development stages of the ‘M8’ cultivar but was low only in the first to third developmental stages of the ‘M3’ cultivar, being high in the late seed developmental stage. Its target gene, *CNR2* (comp59426_c0), showed low expression in the late seed developmental stage, but the highest expression in the third seed developmental stage in the ‘M3’ cultivar; however, the expression of *CNR2* was the lowest in the third seed developmental stage in the ‘M8’ cultivar. 

For ath-miR858b, its expression was lower in the developing seeds in the ‘M8’ cultivar than in the ‘M3’ cultivar. The expression of ath-miR858b was the highest in the early seed developmental stage in the ‘M3’ cultivar, and then gradually decreased; however, its lowest expression in the ‘M8’ cultivar appeared in the third developmental stage, and then gradually increased. The expression of its two target genes, *MYB82* (comp278917_c0) and *MYB3* (comp65157_c0), showed a general decline in the developing seeds of the ‘M3’ and ‘M8’ cultivars; however, *MYB3* showed the lowest expression in the third seed developmental stage, and then slightly increased. In the third seed developmental stage, the expression of *MYB3* was lower in ‘M3’ than in ‘M8’, and its rate of increase in ‘M3’ was slower than that in ‘M8’ from the third to the fourth seed developmental stages. In addition, the expression of *MYB44* (comp31235_c0), another target of ath-miR858b, showed a slight difference between the ‘M3’ and ‘M8’ cultivars. In the ‘M8’ cultivar, *MYB44* consistently maintained high expression in all the seed development stages, but in the ‘M3’ cultivar, its expression was the highest in the early seed developmental stage, and the lowest in the second seed developmental stage, before it gradually increased. 

The expression of hpe-miR162a_L-2 was low in the developing seeds of the ‘M3’ cultivar, but high in the early seed developmental stage, and then it gradually decreased in the ‘M8’ cultivar. Its target gene, *ARF19* (comp57222_c0), had a similar expression pattern in the developing seeds of the ‘M3’ and ‘M8’ cultivars; its expression was high in the early seed developmental stage, but gradually decreased in the second seed developmental stage, and then its expression increased in the third seed developmental stage, and maintained the same levels from the third to fourth seed developmental stages.

### 3.7. MiRNA–mRNA Regulatory Module Involved in Growth and Development

According to the target prediction annotation information, 12 key miRNAs and their target genes related to growth and development (Appendix A) were identified in the high- and low-oil cultivars of tea oil camellia. On the basis of the heat map analysis, the genes targeted by relevant miRNAs were ubiquitously expressed in the developing seeds of tea oil camellia, and the different miRNA expression levels were significantly different between the ‘M3’ and ‘M8’ cultivars (Table 3 and Figure 10). On the one hand, the expression pattern of han-miR156a_L+1 was similar in the ‘M3’ and ‘M8’ cultivars, with significantly low expression levels in the early seed developmental stage, that sharply increased to the highest levels in the late seed developmental stage. Its two target genes, *SPL4* (comp60372_c0) and *SBP2* (comp62492_c0), had the highest expression in the early seed developmental stage in the two cultivars, and the lowest in the late developmental stage. The expression pattern of rco-miR156e_L+1R-1 was similar to that of han-miR156a_L+1, but the expression patterns of its three target genes, *ACS1* (comp66521_c0), *SBP2* (comp62492_c0), and *SBP1* (comp61590_c1), showed a downward trend in the developing seeds of the two cultivars.

On the other hand, the expression of some, such as mtr-miR156h-3p_1ss8AC and col-miRn-3p-9080_426, showed significant differences between the ‘M3’ and ‘M8’ cultivars. For col-miRn-3p-9080_426, its expression was gradually upregulated in the ‘M8’ cultivar from the early to third seed developmental stages but was the lowest in the fourth seed developmental stage. By contrast, col-miRn-3p-9080_426 consistently showed higher expression in all the seed developmental stages in the ‘M3’ cultivar than in the ‘M8’ cultivar. Its one target gene, *COL13* (comp63752_c0), was downregulated in the two cultivars in the late seed developmental stage. The expression pattern of another target gene, *ERF115* (comp58165_c0), was similar to that of the *COL13* gene in the ‘M3’ cultivar, but in the ‘M8’ cultivar, the expression of *ERF115* was the highest in the early seed developmental stage and decreased rapidly to a low level in the third seed developmental stage; after this, it showed a slight increase in the late seed developmental stage. For mtr-miR156h-3p_1ss8AC, its expression was significantly lower in the ‘M3’ cultivar than in the ‘M8’ cultivar in the early developmental stage, and it increased slowly after the early seed developmental stage in the ‘M3’ cultivar; however, it was consistently high throughout all the seed developmental stages in the ‘M8’ cultivar. *TCP24* (comp65957_c0), a target gene of miR156h-3p_1ss8AC, had high expression in the ‘M3’ cultivar in the early seed developmental stage, but it was consistently low in the ‘M8’ cultivar in all the seed developmental stages.

According to these results, these identified miRNAs and their target genes show highly correlative relationships between their expression levels during seed development, and the differences in the expression levels between the two cultivars, or among the different seed developmental stages, might exert a great influence on the growth and development of tea oil camellia seeds.

### 3.8. MiRNA–mRNA Regulatory Modules Involved in Resistance, Yield, and Quality

According to the enrichment analysis of the target genes of the differentially expressed miRNAs, 11 miRNA–mRNA regulatory modules related to stress resistance were identified, mainly involving two categories: abiotic stresses and biotic stresses (Table 4 and Appendix A). In addition, two miRNA–mRNA regulatory modules, which may affect the yield and quality of tea oil camellia seed, were also predicted (Table 4 and Appendix A). Their expression patterns and levels appeared to be obviously different between the ‘M3’ and ‘M8’ cultivars, and among different seed developmental stages (Figure 11). On the one hand, the expression of han-miR156a_L 1 in the ‘M3’ and ‘M8’ cultivars was the lowest in the early seed developmental stage, and then increased gradually; accordingly, its target gene, *SPL16* (comp56544_c0), showed the highest expression in the early seed developmental stage, and then decreased in these two cultivars. 

On the other hand, the expression patterns of ath-miR858b and nta-MIR6149a-p5_2ss18CT21TA were different in the high- and low-oil cultivars. Their expression was consistently high in the ‘M3’ cultivar throughout the seed developmental stages, but for the ‘M8’ cultivar, the expression was high in the early and second seed developmental stages, and then gradually decreased to the lowest value in the third seed developmental stage, and then slowly increased. Accordingly, the expression of *MYB4* (comp31687_c0), a target gene of ath-miR858b, was the lowest in the ‘M8’ cultivar in the fourth seed developmental stage. The expression of *CPR30* (comp65951_c0), a target gene of nta-MIR6149a-p5_2ss18CT21TA, was the lowest in the ‘M3’ cultivar in the second seed developmental stage, but its highest expression in the ‘M8’ cultivar appeared in the third seed developmental stage.

For ppe-miR172a-5p_1ss21GA, its expression remained high in the ‘M3’ cultivar in all the seed developmental stages, but in the ‘M8’ cultivar, its expression was only high in the early seed developmental stage, and then significantly decreased and reached its lowest expression level in the fourth seed developmental stage. Its target gene, *PAB2*, showed the lowest expression in the ‘M3’ cultivar in the late seed developmental stage, and its highest expression appeared in the early seed developmental stage in the ‘M8’ cultivar.

For mtr-miR156h-3p_1ss8AC, its expression remained consistently high in the ‘M8’ cultivar throughout the four different seed developmental stages, but in the ‘M3’ cultivar, its expression was the lowest in the early seed developmental stage, and then gradually increased. However, its two target genes, *BAG7* (comp63682_c0) and *NRT1.7* (comp66713_c0), showed low expression in the ‘M3’ cultivar in the early seed developmental stage, which then gradually decreased to the lowest level in the late seed developmental stage. Although the expression of these two target genes was also the lowest in the ‘M8’ cultivar in the late seed developmental stage, it was still higher than that in the ‘M3’ cultivar.

### 3.9. Validation of Expression Levels of miRNAs and Its Target mRNAs by qRT-PCR

To validate the high-throughput sequencing data, six regulatory modules of differentially expressed miRNAs and their target genes were selected, including han-miR156a_L+1-*SPL4*, rco-miR156e_L+1R-1-*SBP1*, mtr-miR156h-3p_1ss8AC-*TCP24*, ppe-miR172a-5p_1ss21GA-*PAB2*, nta-MIR6149a-p5_2ss18CT21TA-*CPR30,* and col-miRn-3p-9080_426-*ERF115*. Their expression patterns and levels in the high- and low-oil cultivars of tea oil camellia at the four different seed developmental stages were identified by qRT-PCR analysis (Figure 12). The results show that six miRNA–mRNA regulatory modules had differential expression, and the expression patterns of most of the miRNAs and their target genes were consistent with the results obtained in the high-throughput small RNA sequencing data. All six selected miRNAs showed opposite expression trends relative to their target genes.

Han-miR156a_L 1 was upregulated in high- and low-oil cultivars of tea oil camellia, and its expression was the lowest in the early seed developmental stage, and highest in the late developmental stage (Figure 12a). The expression of han-miR156a_L 1 in the ‘M3’ cultivar was higher than that in the ‘M8’ cultivar; by contrast, the expression of its target gene, *SPL4*, was the highest in the early seed developmental stage, and then decreased sharply to the lowest level in the late seed developmental stage (Figure 12a). The expression of rco-miR156e_L 1R-1 was upregulated in the ‘M8’ cultivar; it gradually increased from the early to third seed developmental stages, peaking in the third seed developmental stage, and then decreasing slightly (Figure 12b). The expression of rco-miR156e_L 1R-1 in the ‘M3’ cultivar was upregulated continuously and reached the highest value in the late seed developmental stage, but always remained lower than in the ‘M8’ cultivar (Figure 12a). Interestingly, the target gene of rco-miR156e_L 1R-1, *SBP1*, was downregulated in the ‘M3’ and ‘M8’ cultivars, and its expression was the highest in the early seed developmental stage, and lowest in the late developmental stage (Figure 12b).

The expression of mtr-miR156h-3p_1ss8AC was upregulated in the ‘M3’ cultivar, and was significantly lower than that in the ‘M8’ cultivar in the early seed developmental stage; after this, it showed a rapidly increasing trend, with the highest expression in the late seed developmental stage (Figure 12c). By contrast, the expression of mtr-miR156h-3p_1ss8AC in the ‘M8’ cultivar was relatively high in all the seed developmental stages. Surprisingly, the expression of the target gene of mtr-miR156h-3p_1ss8AC, *TCP24*, in the ‘M3’ cultivar, was also low in the early seed developmental stage (Figure 12c), but was then rapidly upregulated; however, its expression in the ‘M8’ cultivar remained low in all the seed developmental stages (Figure 12c). The expression of miR156h-3p_1ss8AC was found to be negatively correlated with the expression of its target gene in the low-oil ‘M8’ cultivar.

The expression of ppe-miR172a-5p_1ss21GA was the highest in the ‘M3’ cultivar in the second seed developmental stage (Figure 12d), while the expression of its target gene, *PAB2*, was lower in the second seed developmental stage than in the early and third seed developmental stages. The expression pattern of ppe-miR172a-5p_1ss21GA was similar to that of its target gene in the ‘M8’ cultivar, and both were first upregulated and then downregulated (Figure 12d). The expression of ppe-miR172a-5p_1ss21GA was found to be negatively correlated with the expression of its target gene in the high-oil ‘M3’ cultivar.

Nta-MIR6149a-p5_2ss18CT21TA had the highest expression in the ‘M3’ cultivar in the third seed developmental stage, but its expression was the lowest in the ‘M8’ cultivar in the third seed developmental stage and was significantly lower than that in the ‘M3’ cultivar (Figure 12e). Its target gene, *CPR30*, was low in the ‘M3’ cultivar in the fourth developmental stage. The expression of the *CPR30* gene in the ‘M3’ cultivar was higher than that in the ‘M8’ cultivar in the early seed developmental stage but was lower in the ‘M3’ cultivar than in the ‘M8’ cultivar in the other three seed developmental stages (Figure 12e). The expression of col-miRn-3p-9080_426 peaked in the ‘M3’ cultivar in the second seed developmental stage, and in the ‘M8’ cultivar in the third seed developmental stage (Figure 12f). It was first upregulated and then downregulated in the ‘M3’ cultivar. The target gene, *ERF115*, of nta-MIR6149a-p5_2ss18CT21TA, showed the lowest expression in the ‘M8’ cultivar in the third seed developmental stage, and its expression first increased and then decreased in the low-oil ‘M3’ cultivar (Figure 12f).

The above results, obtained by qRT-PCR validation, are consistent with the sequencing results for the expression patterns and levels of miRNAs and their targeted genes (Figure 12), indicating the accuracy and high efficiency of the small RNA high-throughput sequencing results, and reflecting the dependability of the qRT-PCR method used to identify the temporal and spatial expression characteristics and expression differences of related functional miRNAs and target genes in tea oil camellia at different seed developmental stages.

### 3.10. Validation of Targeting Relationship of the cpa-miR393_R-1-AFB2 Regulatory Module by Luciferase Assays

A target prediction analysis server, psRNA target [56], was used to assess the complementarity between cpa-miR393_R-1 and the target site of *AFB2*, with the default parameters and a maximum expectation value of 4 (the number of mismatches allowed). The potential of cpa-miR393_R-1 to target the AFB2 3′-UTR was predicted (Figure 13a). The luciferase activity in the HEK-293T cells cotransfected with the cpa-miR393_R-1 recombinant expression vector, and the expression vector containing the 3′-UTR of *AFB2* fused with the reporter gene was decreased by nearly 36.26% (Student’s *t*-test, *p* < 0.05) compared to that in the control group (Figure 13b). These results indicate that the *AFB2* gene was targeted by cpa-miR393_R-1.

## 4. Discussion

### 4.1. Identification of miRNAs in Developing Seeds of Tea Oil Camellia

In this study, a total of 196 miRNAs, including 156 known and 40 novel miRNAs, were identified in the developing seeds of the ‘M3’ and ‘M8’ cultivars of tea oil camellia. For the 196 miRNAs identified in the seeds of tea oil camellia, a total of 17,166 target genes were predicted, and there were multiple target genes targeted by the same miRNA, and the same gene was targeted by several miRNAs. This is consistent with the report that one miRNA can regulate two or more target genes [57], and that the same target gene can also be targeted by different miRNAs [58]. The 196 miRNAs identified in tea oil camellia seeds appeared most frequently in the gma species; most of the conserved miRNAs were from 35 miRNA families, such as miR160, miR156, miR159, miR162, and miR164. This also demonstrates that miRNAs are highly conserved in their evolutionary relationships, especially in plants [59]. 

The exploration of the molecular mechanisms of miRNA regulation involved in various biological pathways in plants is a field of great interest to current researchers [60]. A total of 33,418 miRNA–mRNA regulatory modules were predicted in this study, 25,988 of which were related to different biological processes, including 21,132 miRNA–mRNA modules involved in different cell components, and 4675 modules involved in molecular function. In these, 107 significantly enriched GO terms were related to the nucleus, sequence-specific DNA binding, the regulation of transcription, the DNA-template, and electron carrier activity. In addition, 31 significantly enriched KEGG pathways were mainly involved in pyruvate metabolism, the regulation of the actin cytoskeleton, fatty acid biosynthesis, and the citrate cycle. These results indicate that miRNAs regulate different biological functions and metabolic pathways in the developing seeds of tea oil camellia. Excavating these important miRNA–mRNA regulatory modules could enable a clear understanding of the biological functions and metabolic pathways of miRNA-regulated target genes during seed development in this species, providing favorable resources for improving tea oil camellia in the future.

### 4.2. MiRNA–mRNA Modules Involved in Lipid Metabolism in Tea Oil Camellia

Studies have shown that miRNAs are involved in lipid metabolism by regulating targeted genes in oil crops, such as *B. napus* [61], hazel [36], oil palm [37], olive [38], peony [22], walnut [10], and yellowhorn [11]. In this study, the mtr-miR156h-3p_1ss8AC had high expression in the developing seeds of tea oil camellia and targeted the *fabG* gene. The *fabG* gene encodes a member of the ketoacyl reductase family of proteins, and it is an essential enzyme for type II fatty acid biosynthesis [62]. This is consistent with miR156 influencing the lipid biosynthesis of the seeds in sea buckthorn [24] and rapeseed [63]. 

A total of 10 miRNA–mRNA regulatory modules related to lipid metabolism pathways in the seeds of tea oil camellia were first found in this study (Figure 14). The regulation module of tcc-miR162–*ACC1* may be related to the oil and seed yield of tea oil camellia. This is because *ACCase* is a rate-limiting enzyme that controls the flow of carbon to fatty acid synthesis; not only is it a key gene for lipid and fatty acid biosynthesis in tea oil camellia [64], but its high expression is also closely related to the high yield and oil content of oil palm [65,66]. The *GAPN* gene is related to lipid accumulation [67]; thus, the gma-MIR5368-p3_1ss21CT–*GAPN* regulatory module is involved in lipid metabolism in tea oil camellia.

The remaining regulation modules of csi-miR166e-5p–*KAS1**/S-ACP*-*DES6*, rgl-miR5139_L-1-*KAS3B*, cpa-miR164d-*Mcat*, col-miRn-5p-21064_221-*FATB1*, aly-miR393a-3p_1ss12TC -*MOD1,* and ath-miR172a-∆*9D* were related to the fatty acid biosynthesis metabolic pathway, because these target genes, such as *KASIII* [68], *Mcat* [69], *MOD1* [70], and ∆*9D* [24], reported in previous studies (Appendix A), are related to lipid biosynthesis and fatty acid formation and accumulation. *KASI* is involved in the formation of C16 or C18 saturated fatty acids catalyzed by the fatty acid synthase complex [71]. *S-ACP*-*DES6* (comp67006_c0) regulates the contents of oleic acid in the seeds of tea oil camellia. The expression of csi-miR166e-5p was similarly upregulated in the ‘M3’ and ‘M8’ cultivars, and its target gene, *S-ACP*-*DES6*, had a trend of upregulation that promoted the formation and accumulation of oleic acid in developing seeds. This is consistent with the result from a previous report on the *S-ACP*-*DES* gene, stating that it is a key gene involved in oleic acid biosynthesis in tea oil camellia [72] and *Arabidopsis* [73]. Previous studies found that the overexpression of the *FATB1* gene in *Arabidopsis* seeds resulted in a four-fold increase in the palmitic acid content in the seed oil [74]. This indicates that the regulation module of col-miRn-5p-21064_221–*FATB1* in tea oil camellia may be related to fatty acid formation and accumulation in its seed oil. 

The analysis of the expression patterns and expression levels for the miRNA–mRNA regulatory modules involved in lipid metabolism provides a clear perspective for understanding the role of miRNAs in the lipid biosynthesis of tea oil camellia seeds, indicating the potential for the utilization of miRNAs to improve the seed oil content and oil components in tea oil camellia in the future.

### 4.3. MiRNA–mRNA Regulatory Modules Involved in Seed Size in Tea Oil Camellia

In this study, our data show that some target genes in the 23 miRNA–mRNA regulatory modules in the seeds of tea oil camellia were transcription factors (e.g., ARF, CNR, MYB, and MED), which are related to the seed size (Appendix A). This is consistent with most of the target genes encoding transcription factors in miRNA–mRNA modules related to seed size [18,75], mainly including ARF [24,29], CNR [76], and MYB [77]. In chickpea and oilseed crops, transcription factors of the MYB and ARF families are involved in seed size/weight determination [29,30]. Negative regulatory modules of hpe-miR162a_L-2–*ARF19* may be involved in the seed development of tea oil camellia because the low expression of hpe-miR162a_L-2 and the high expression of *ARF19* in the early seed development stage are helpful for auxin biosynthesis. The function of SlMYB is as a candidate gene involved in the fruit set process in tomatoes [78]. In tea oil camellia, ath-miR858b may regulate seed development by targeting MYB transcription factors. In the developing seeds of tea oil camellia, the three transcription factors, *MYB82* (comp278917_c0), *MYB3* (comp65157_c0), and *MYB44* (comp31235_c0), cotargeted by ath-miR858b, showed a declining trend and lower expression in the two cultivars of ‘M3’ and ‘M8’, and the seed size gradually increased in the developing seeds. This is consistent with the evidence that the transcription factor, MYB89, represses the seed yield in *Arabidopsis* [79]. 

### 4.4. MiRNA-Target Genes Involved in Growth, Development, and Resistance in Tea Oil Camellia

In this study, 12 miRNA–mRNA regulatory modules related to the growth and development of tea oil camellia seed were identified because these targeted the gene transcription factors, ACS1, AFB2, COL13, ERF115, SBP1, SBP2, SCL16, SPL4, T1R1, and TCP24, which are related to growth and development (Appendix A). For example, the *T**CP*24 modulates secondary cell-wall thickening and anther endothecium development [80]; the *Lc-ACS1* gene might be the key gene involved in fruitlet abscission in litchis [81]; *SBP1* is an important component of the atypical SCF complex, playing a unique role in the self-compatibility system of gametes [82]. An SBP2–GFP fusion protein under the control of the GmSBP2 promoter accumulates in the vascular tissues of vegetative organs, which is consistent with the proposed involvement of *SBP* in sucrose-transport-dependent physiological processes [83].

Han-miR156a_L+1 showed significantly low expression in the early seed developmental stage in the ‘M3’ and ‘M8’ cultivars of tea oil camellia, and then increased sharply to its highest level in the late seed developmental stage. By contrast, its two target genes, *SPL4* and *SBP2*, had the highest expression in the early seed developmental stage in two cultivars, and the lowest expression appeared in the late developmental stage. This indicated that the negative regulation modules of han-miR156a_L+1–*SPL4**/SBP2* promote seed development in the early seed developmental stage in tea oil camellia. This is consistent with miR156 participating in the early morphogenesis of *Arabidopsis* embryos by regulating the targets *SPL10* and *SPL11* [33], and it may also mediate the targeted regulation of *SBP* expression levels, affecting the flowering time in plants [29]. 

In tea oil camellia seeds, ccl-miR171 and ssl-miR171b_2ss11CT19AT may regulate seed development by targeting the *SCL16* gene. miRNA171 controls the growth and development of flowers, roots, and stems by targeting SCL transcription factor family members in *Arabidopsis* [84]. The genes of *AFB2* and *T1R1* are auxin-receptor genes regulated by miR393, which is involved in hormone regulation and signal transduction in plants [48]. Thus, two regulation modules of cpa-miR393_R-1–AFB2/T1R1 in tea oil camellia may be related to seed development. 

In addition, 13 miRNA–mRNA regulatory modules were related to resistance and yield in the seeds of tea oil camellia (Appendix A), in which the target genes of *APUM*5 [85], *BAG*7 [86], *CPR*30 [87], *ERF*106 [88], *MYB*4 [89], *Os03g0214200* [90], *SGS*3 [91], and *SPL*16 were [92] involved in resistance, and *NRT*1.7 was related to yield [93]. 

## 5. Conclusions

In conclusion, a total of 196 miRNAs, including 156 known miRNAs from 35 families, and 40 novel miRNAs, were identified in the developing seeds of high- and low-oil two cultivars of tea oil camellia, in which 55 significantly differentially expressed miRNAs were found, with 34 upregulated miRNAs, and 21 downregulated miRNAs. A total of 9 miRNA–mRNA regulatory modules were related to lipid metabolism. For example, the negative regulatory module of ath-miR858b–*MYB82**/MYB3**/MYB44* represses seed oil biosynthesis, and a regulatory module of csi-miR166e-5p–*S-ACP*-*DES6* (comp67006_c0) was related to the formation and accumulation of oleic acid. A total of 23 miRNA–mRNA regulatory modules were involved in the regulation of seed development, such as the negative regulatory modules of hpe-miR162a_L-2–*ARF19*, involved in early seed development. Twelve miRNA–mRNA regulatory modules regulating growth and development were identified, such as the negative regulatory modules of han-miR156a_L+1–*SPL4**/SBP2*, which promote early seed development.

## Figures and Tables

**Figure 1 cells-11-00071-f001:**
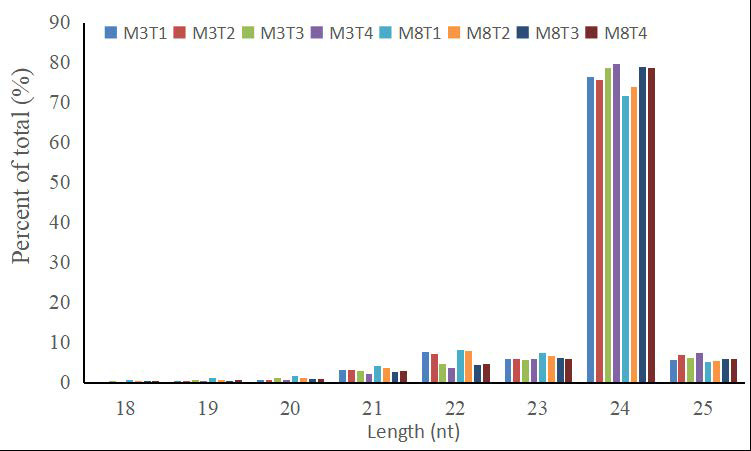
Length distributions of unique miRNAs from the seeds of the ‘M3’ and ‘M8’ cultivars of tea oil camellia at four different developmental stages.

**Figure 2 cells-11-00071-f002:**
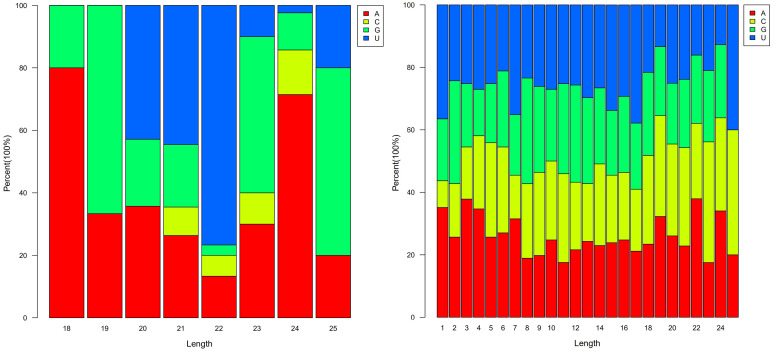
(**a**) First nucleotide, and (**b**) total nucleotide biases of tea oil camellia miRNA.

**Figure 3 cells-11-00071-f003:**
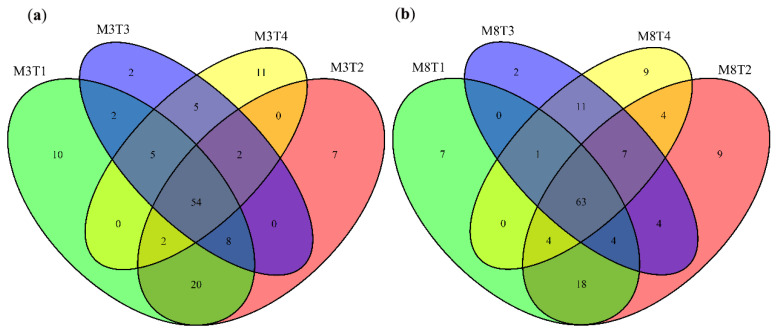
Venn diagrams of the detected miRNAs among (**a**) the seeds of ‘M3’, and (**b**) the seeds of ‘M8’, at four different developmental stages.

**Figure 4 cells-11-00071-f004:**
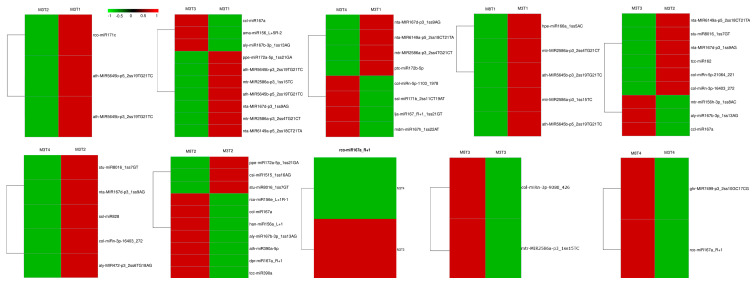
Heatmap of differentially expressed miRNAs among different groups.

**Figure 5 cells-11-00071-f005:**
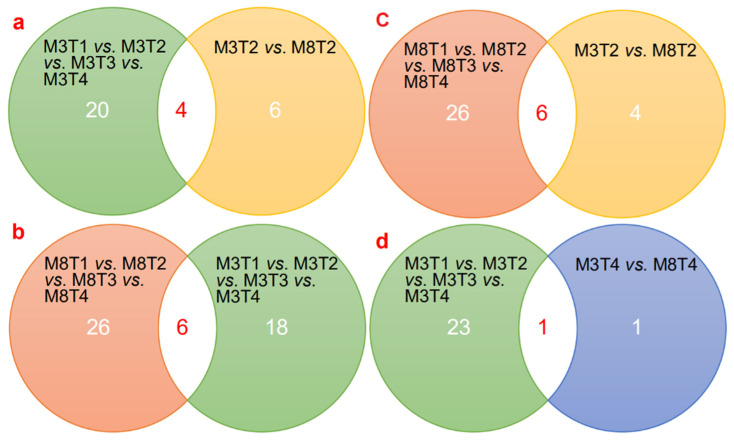
Venn diagrams of differentially expressed miRNAs between different comparison groups. (**a**) Distribution of a Venn diagram of differentially expressed miRNAs in M3T1 vs. M3T2 vs. M3T3 vs. M4T4 compared to M3T2 vs. M8T2, and M8T1 vs. M8T2 vs. M8T3 vs. M8T4 compared to M3T2 vs. M8T4; (**b**) Distribution of a Venn diagram of differentially expressed miRNAs in M3T1 vs. M3T2 vs. M3T3 vs. M4T4 compared to M8T1 vs. M8T2 vs. M8T3 vs. M8T4; (**c**) Distribution of a Venn diagram of differentially expressed miRNAs in M8T1 vs. M8T2 vs. M8T3 vs. M8T4 compared to M3T2 vs. M8T2; and (**d**) Distribution of a Venn diagram of differentially expressed miRNAs in M3T1 vs. M3T2 vs. M3T3 vs. M4T4 compared to M3T4 vs. M8T4.

**Figure 6 cells-11-00071-f006:**
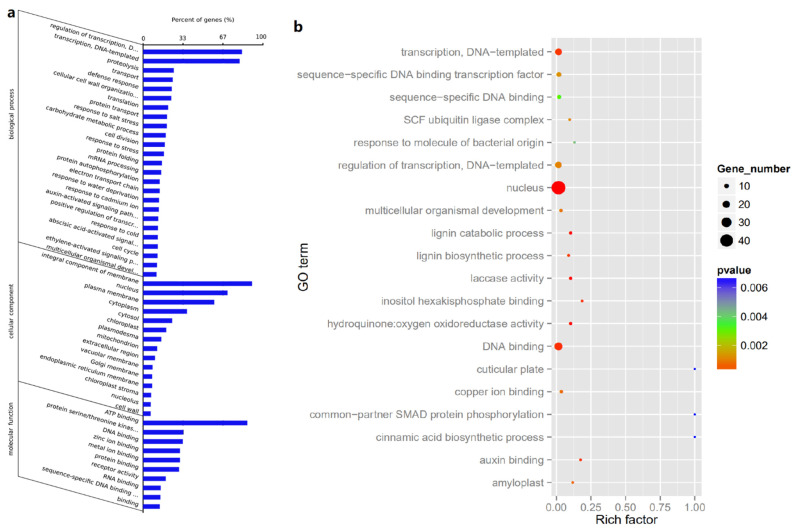
(**a**) GO categories and distribution, and (**b**) scatterplot distribution map of genes targeted by identified miRNAs in the seeds of tea oil camellia.

**Figure 7 cells-11-00071-f007:**
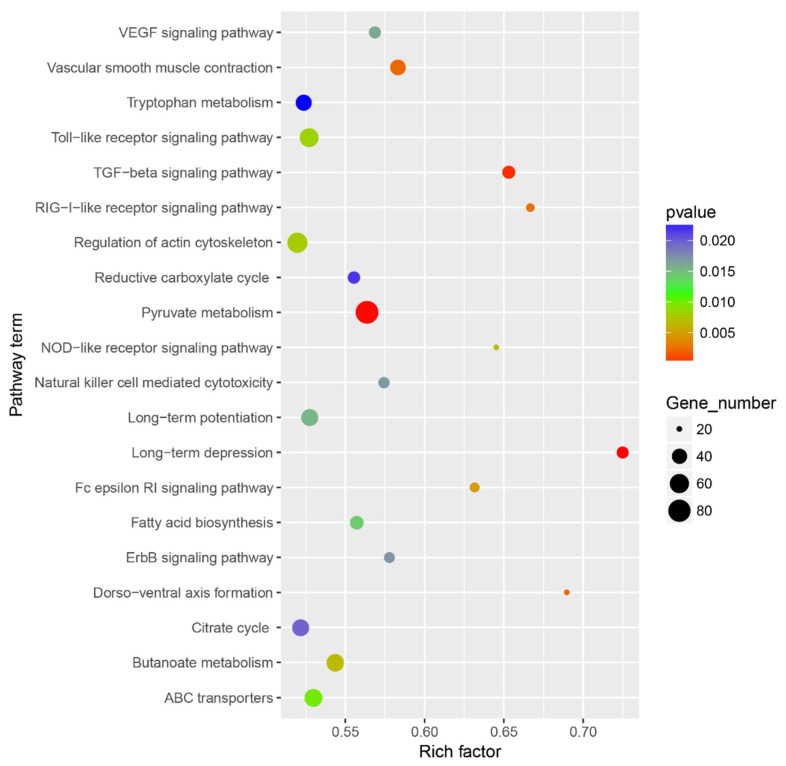
Analyses of significant KEGG enrichment scatterplot distribution map based on miRNA targets.

**Figure 8 cells-11-00071-f008:**
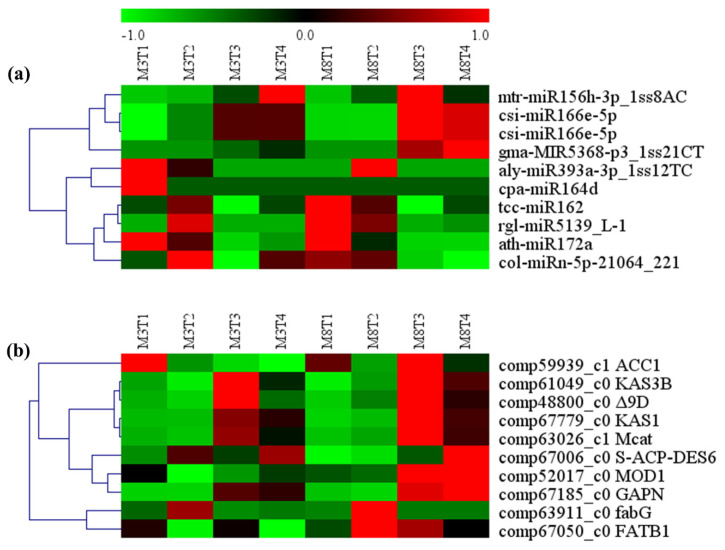
Heat map analysis of (**a**) predicted differentially expressed miRNAs, and (**b**) their target genes, involved in lipid metabolism in the seeds of the ‘M3’ and ‘M8’ cultivars at four developmental stages. Red and green colors indicate high levels and low levels, respectively, of the miRNAs and their targets.

**Figure 9 cells-11-00071-f009:**
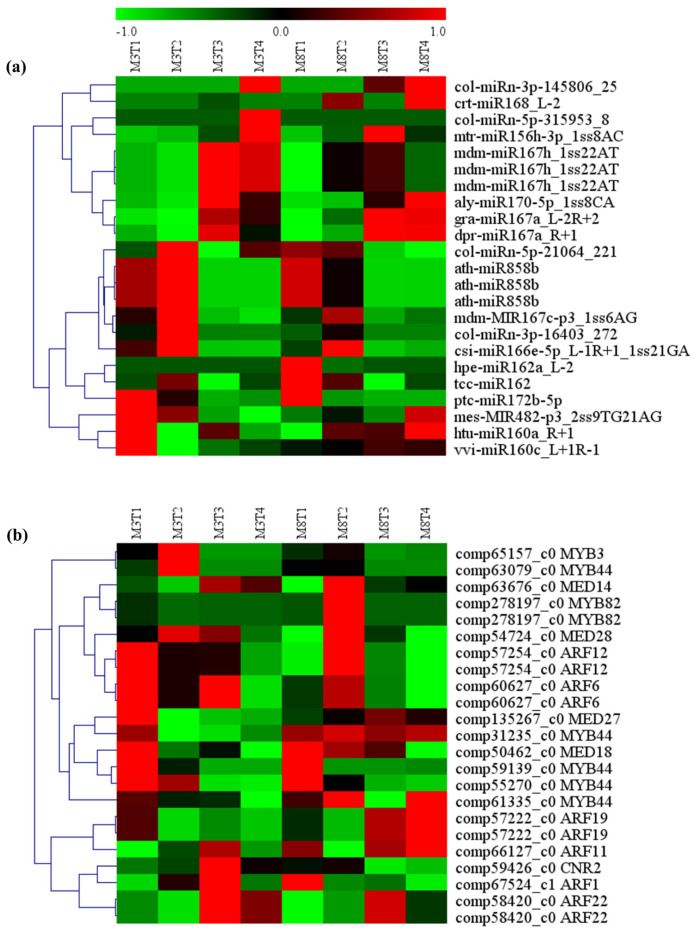
Heat map analysis of (**a**) predicted differentially expressed miRNAs, and (**b**) their target genes, involved in seed development from seeds of the ‘M3’ and ‘M8’ cultivars at four developmental stages. Red and green colors indicate high levels and low levels, respectively, of the miRNAs and their targets.

**Figure 10 cells-11-00071-f010:**
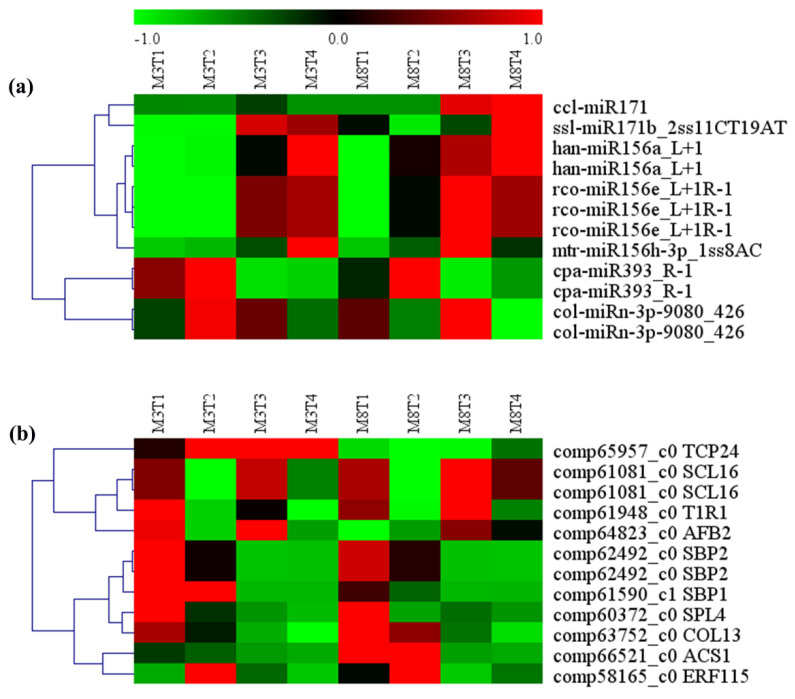
Heat map analysis of (**a**) predicted differentially expressed miRNAs, and (**b**) their target genes, involved in the growth and development of seeds from the ‘M3’ and ‘M8’ cultivars at four developmental stages. Red and green colors indicate high levels and low levels, respectively, of the miRNAs and their targets.

**Figure 11 cells-11-00071-f011:**
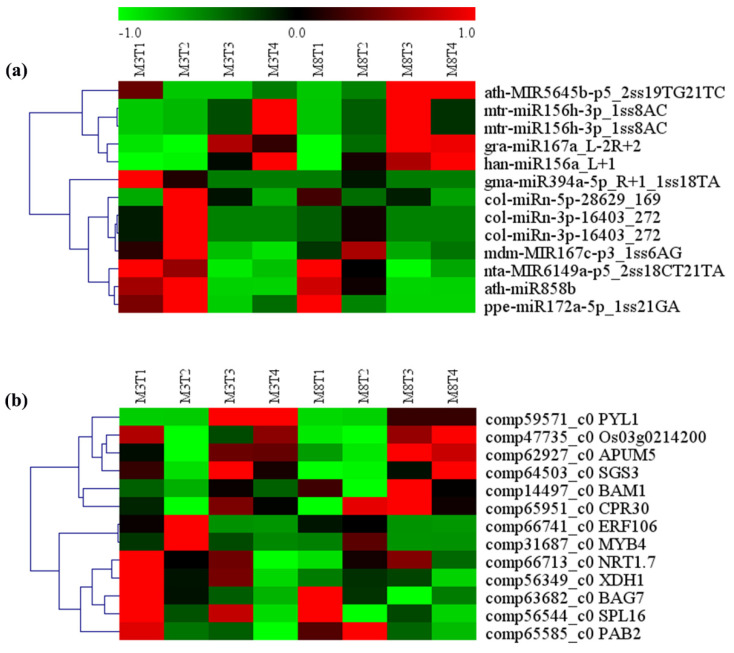
Heat map analysis of (**a**) predicted differentially expressed miRNAs, and (**b**) their target genes, related to resistance, yield, and quality from the seeds of the ‘M3’ and ‘M8’ cultivars at four developmental stages. Red and green colors indicate high levels and low levels, respectively, of the miRNAs and their targets.

**Figure 12 cells-11-00071-f012:**
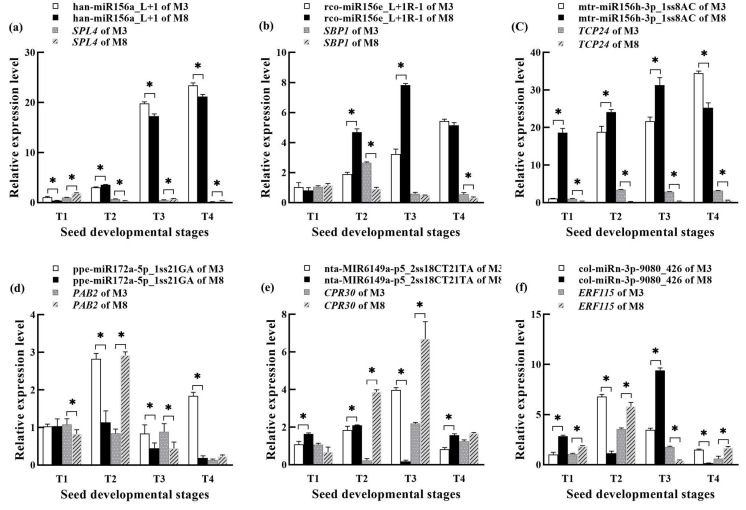
Expression differences of selected miRNA–mRNA pairs in the seeds of the ‘M3’ and ‘M8’ cultivars of tea oil camellia at four different developmental stages: (**a**) han-miR156a_L+1-*SPL4*; (**b**) rco-miR156e_L+1R-1-*SBP1*; (**c**) mtr-miR156h-3p_1ss8AC-*TCP24*; (**d**) ppe-miR172a-5p_1ss21GA-*PAB2*; (**e**) nta-MIR6149a-p5_2ss18CT21TA-*CPR30*; and (**f**) col-miRn-3p-9080_426-*ERF115*. The error bars indicate standard deviations for the three biological replicates. * represents significant differences between the two cultivars at the same seed developmental stage, on the basis of a Student’s *t*-test at *p* < 0.05.

**Figure 13 cells-11-00071-f013:**
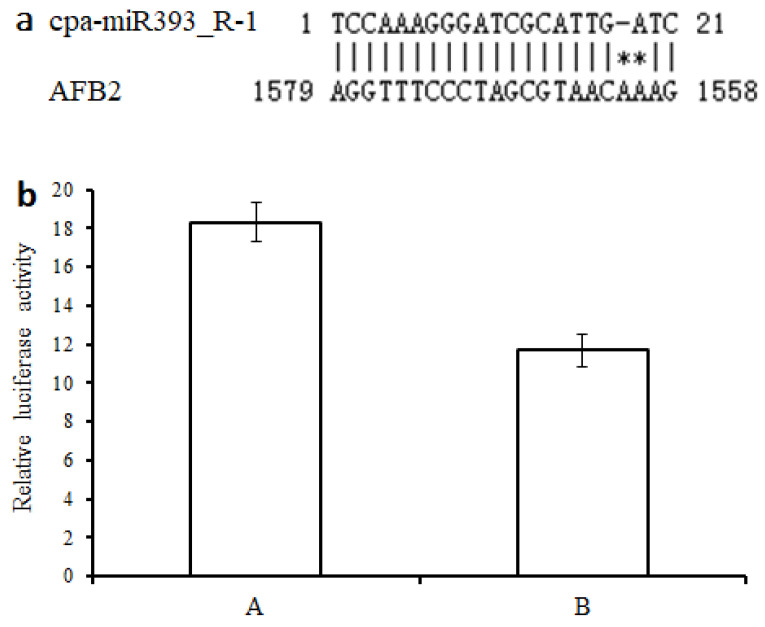
The validation of cpa-miR393_R-1 and target AFB2: (**a**) prediction of the binding sites of cpa-miR393_R-1 in AFB2 mRNA using psRNATarget; (**b**) effect of cpa-miR393_R-1 expression on luciferase activity in transfected cells: (A) pCDNA3.1 + pmirGLO-AFB2, and (B) pCDNA3.1- cpa-miR393_R-1 + pmirGLO-AFB2. Data are represented as the means ± standard deviations (SDs) from three independent experiments.

**Figure 14 cells-11-00071-f014:**
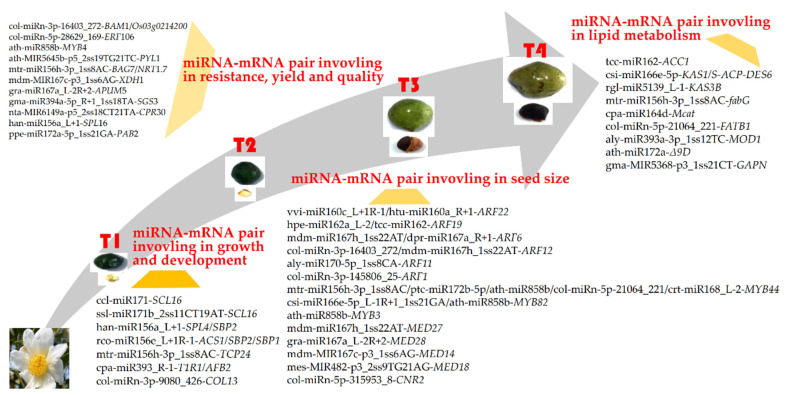
miRNA–mRNA regulatory modules in developing seeds of tea oil camellia.

**Table 1 cells-11-00071-t001:** Predicted miRNAs and their target genes involved in lipid metabolism.

miRNA ID	Gene ID	Gene Name	Gene Annotation	KEGG Pathway
tcc-miR162	comp59939_c1	*ACC1*	acetyl-CoA carboxylase/biotin carboxylase	Pyruvate metabolism
csi-miR166e-5p	comp67779_c0	*KAS1*	3-oxoacyl-[acyl-carrier-protein] synthase I	
csi-miR166e-5p	comp67006_c0	*S-ACP-DES6*	acyl-[acyl-carrier-protein]-desaturase	
rgl-miR5139_L-1	comp61049_c0	*KAS3B*	3-oxoacyl-[acyl-carrier-protein] synthase III	
mtr-miR156h-3p_1ss8AC	comp63911_c0	*fabG*	3-oxoacyl-[acyl-carrier protein] reductase	
cpa-miR164d	comp63026_c1	*Mcat*	[acyl-carrier-protein] S-malonyltransferase	Fatty acid biosynthesis metabolic
col-miRn-5p-21064_221	comp67050_c0	*FATB1*	fatty acyl-ACP thioesterase B	
aly-miR393a-3p_1ss12TC	comp52017_c0	*MOD1*	enoyl-[acyl-carrier protein] reductase I	
ath-miR172a	comp48800_c0	*Δ9D*	acyl-[acyl-carrier-protein] desaturase	
gma-MIR5368-p3_1ss21CT	comp67185_c0	*GAPN*	glyceraldehyde-3-phosphate dehydrogenase (NADP)	Glycolysis/gluconeogenesis

**Table 2 cells-11-00071-t002:** Predicted miRNAs and their target genes involved in seed size.

miRNA ID	Gene ID	Gene Name	Gene Annotation
vvi-miR160c_L+1R-1	comp58420_c0	*ARF22*	auxin response factor 22
htu-miR160a_R+1	comp58420_c0	*ARF22*	auxin response factor 22
hpe-miR162a_L-2	comp57222_c0	*ARF19*	auxin response factor 19
tcc-miR162	comp57222_c0	*ARF19*	auxin response factor 19
mdm-miR167h_1ss22AT	comp60627_c0	*ARF6*	auxin response factor 6
dpr-miR167a_R+1	comp60627_c0	*ARF6*	auxin response factor 6
col-miRn-3p-16403_272	comp57254_c0	*ARF12*	auxin response factor 12
mdm-miR167h_1ss22AT	comp57254_c0	*ARF12*	auxin response factor 12
aly-miR170-5p_1ss8CA	comp66127_c0	*ARF11*	auxin response factor 11
col-miRn-3p-145806_25	comp67524_c1	*ARF1*	auxin response factor 1
mtr-miR156h-3p_1ss8AC	comp55270_c0	*MYB44*	Transcription factor MYB44
ptc-miR172b-5p	comp61335_c0	*MYB44*	Transcription factor MYB44
ath-miR858b	comp31235_c0	*MYB44*	Transcription factor MYB44
col-miRn-5p-21064_221	comp63079_c0	*MYB44*	Transcription factor MYB44
crt-miR168_L-2	comp59139_c0	*MYB44*	Transcription factor MYB44
csi-miR166e-5p_L-1R+1_1ss21GA	comp278197_c0	*MYB82*	Transcription factor MYB82
ath-miR858b	comp278197_c0	*MYB82*	Transcription factor MYB82
ath-miR858b	comp65157_c0	*MYB3*	Transcription factor MYB3
mdm-miR167h_1ss22AT	comp135267_c0	*MED27*	Mediator of RNA polymerase II transcription subunit 27
gra-miR167a_L-2R+2	comp54724_c0	*MED28*	Mediator of RNA polymerase II transcription subunit 28
mdm-MIR167c-p3_1ss6AG	comp63676_c0	*MED14*	Mediator of RNA polymerase II transcription subunit 14
mes-MIR482-p3_2ss9TG21AG	comp50462_c0	*MED18*	Mediator of RNA polymerase II transcription subunit 18
col-miRn-5p-315953_8	comp59426_c0	*CNR2*	Cell number regulator 2

**Table 3 cells-11-00071-t003:** Predicted miRNAs and their target genes involved in growth and development.

miRNA ID	Gene ID	Gene Name	Gene Annotation
ccl-miR171	comp61081_c0	*SCL16*	Scarecrow-like protein 6
ssl-miR171b_2ss11CT19AT	comp61081_c0	*SCL16*	Scarecrow-like protein 6
han-miR156a_L+1	comp60372_c0	*SPL4*	Squamosa promoter-binding-like protein 4
han-miR156a_L+1	comp62492_c0	*SBP2*	Squamosa promoter-binding protein 2
rco-miR156e_L+1R-1	comp66521_c0	*ACS1*	1-aminocyclopropane-1-carboxylate synthase
rco-miR156e_L+1R-1	comp62492_c0	*SBP2*	Squamosa promoter-binding protein 2
rco-miR156e_L+1R-1	comp61590_c1	*SBP1*	Squamosa promoter-binding protein 1
mtr-miR156h-3p_1ss8AC	comp65957_c0	*TCP24*	Transcription factor TCP24
cpa-miR393_R-1	comp61948_c0	*T1R1*	transport inhibitor response 1
cpa-miR393_R-1	comp64823_c0	*AFB2*	Protein AUXIN SIGNALING F-BOX 2
col-miRn-3p-9080_426	comp63752_c0	*COL13*	Zinc finger protein CONSTANS-LIKE 13
col-miRn-3p-9080_426	comp58165_c0	*ERF115*	EREBP-like factor

**Table 4 cells-11-00071-t004:** Predicted miRNAs and their target genes involved in resistance, yield, and quality.

miRNA ID	Gene ID	Gene Name	Gene Annotation
col-miRn-3p-16403_272	comp14497_c0	*BAM*1	leucine-rich repeat receptor-like serine/threonine-protein kinase, BAM1
col-miRn-3p-16403_272	comp47735_c0	*Os03g0214200*	Ninja-family protein, Os03g0214200
col-miRn-5p-28629_169	comp66741_c0	*ERF*106	EREBP-like factor
ath-miR858b	comp31687_c0	*MYB*4	Myb-related protein, Myb4
ath-MIR5645b-p5_2ss19TG21TC	comp59571_c0	*PYL*1	Abscisic acid receptor, PYL1
mtr-miR156h-3p_1ss8AC	comp63682_c0	*BAG*7	BAG family molecular chaperone regulator 7
mtr-miR156h-3p_1ss8AC	comp66713_c0	*NRT*1.7	proton-dependent oligopeptide transporter, POT family
mdm-MIR167c-p3_1ss6AG	comp56349_c0	*XDH*1	xanthine dehydrogenase/oxidas
gra-miR167a_L-2R+2	comp62927_c0	*APUM*5	Pumilio homolog 5
gma-miR394a-5p_R+1_1ss18TA	comp64503_c0	*SGS*3	Protein SUPPRESSOR OF GENE SILENCING 3
nta-MIR6149a-p5_2ss18CT21TA	comp65951_c0	*CPR*30	F-box protein CPR30
han-miR156a_L+1	comp56544_c0	*SPL*16	Squamosa promoter-binding-like protein 16
ppe-miR172a-5p_1ss21GA	comp65585_c0	*PAB*2	20S proteasome subunit alpha 2

## Data Availability

Not applicable.

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
