# Peer review of "Identification of miRNA–mRNA Regulatory Modules Involved in Lipid Metabolism and Seed Development in a Woody Oil Tree (Camellia oleifera)"

_cells, 2021, doi:10.3390/cells11010071_

Round 1

Reviewer 1 Report

The authors completely answered all questions that I previously asked. I still don't think that all miRNA-mRNA pairs detected in this study actually regulate seed development and oil accumulation, but the results will gain some insights into miRNA-mediated gene silencing during seed development in Camellia plants.

Author Response

Thanks for your review on this manuscript. The miRNA-mRNA pairs were predicted by bioinformation in most studies, and the functions of these modules were inferred from the functions of target gene annotated in mRNA-seq. Thus, as you said, the results will provide some insights into miRNA-mediated gene silencing during seed development in Camellia plants.

Reviewer 2 Report

This manuscript is a re-submission of a prvious manuscript with the identical title "Identification of miRNA–mRNA regulatory modules involved in lipid metabolism and seed development in a woody oil tree (Camellia oleifera)"

The authors adequately adressed all the points raised by my previous review on this manuscript and improved the overall quality by adding missing information regarding the methodology, give additional background information in the introduction and and adressed critical point in the results and discussion sections of this manuscript.

To my knowledge no novel experimental data was included to strengthen the previous manuscript and as such the authors tried to be more conservative in their interpretation of the data compared to the previous manuscript.

Minor comments:

In line 761 the authors state a p-value. Please indicate the statistical test used for this calculation, similar to its inclusion for the qRT-PCR data.

Author Response

Thanks for your review on the manuscript, in the revised manuscript, we added ‘Student’s T-test, p < 0.05’ to indicate the statistical test used for this calculation. Happy a merry Christmas and a new year to you. Best regards, Chengjiang Ruan

This manuscript is a resubmission of an earlier submission. The following is a list of the peer review reports and author responses from that submission.

Round 1

Reviewer 1 Report

The manuscript describes comprehensive analysis of miRNA-mediated gene silencing during seed development in tea oil camellia.

   The authors performed miRNA-seq expressed in developing seeds of two camellia lines, M3 and M8, with high and low seed oil contents, respectively. Furthermore, they divided a seed developing process into four stages, extracted RNAs from every stages of two lines with 2 replicates in each, and supplied for making small RNA library and high-throughput sequencing. Many differentially expressed miRNAs between 2 lines and among developmental stages were identified. The authors computationally predicted genes putatively targeted by these miRNAs, and found that many of miRNA-mRNA pairs predicted here showed mutually anti-correlated expression, suggesting gene silencing by these miRNAs. Then several target gene candidates were selected, supplied for making cDNA libraries from tea oil camellia RNAs and reconfirming anti-correlated expression pattern by qRT-PCR. This is a descriptive paper of miRNAs and putatively miRNA-targeted genes expressing in developing seeds of tea oil camellia. Finally, they proved that one of miRNA detected (cpa-miR393_R1) has a potential to reduce the expression of LUC gene fused with AFB2 3'-UTR containing a cpa-miR393_R1 target site.

   Though the results are just limited in transcriptome analysis and there is no direct evidence that miRNAs identified in this study are actually involved in in-vivo target mRNA cleavage or translation repression, I think the information will be beneficial for many researchers aiming development of camellia varieties with higher seed oil content.

   However, I have several major and minor questions as follows. Unless the authors answer all questions, I cannot recommend to accept this paper.

  1. The authors describe details of miRNA-seq analysis. In contrast, little information is shown in the manuscript. The gene ID, "comp***", appear in many tables. What do these genes derive from? Furthermore, no complementary relationship between miRNAs detected in this study and their putative target genes, as shown in Fig. 13a. No target sequence and no complementary relationship... how can I validate the results of miRNA-target finding experiments of this ms?

  1. I understand that tremendous numbers of miRNAs and putative target genes were detected here and it is difficult to show them in concise figures. However, the comparison of miRNA expression and target candidate genes on heatmaps (Figs. 7, 8, 9, 10) is a very very hard job, and I eventually gave up to follow up all of them. In addition, descriptions in the Results section for those figs are also too busy to follow up. I recommend to narrow them down and argue the main points more clearly and concisely.

  1. There are few descriptions about analytical conditions in Target Finder (L180-181) and psRNATarget analyses (L692-694). At first, no citation is shown in the text. Second, there is no description about setting of target finding conditions. For example, G:U wobble pairing often occurs in complementary association of miRNA and its target RNA, but I couldn't know if the authors took wobble pairing in care or not, while Target Finder has a function to detect wobble pairing.

  1. I don't think the title is fully supported by the results shown in this ms. Most of annotations of miRNA-targeted genes predicted are based on annotations determined in model plants, and it is unclear whether the genes of tea oil camellia have functions similar to those annotated. This ms only showed anti-correlated expression of miRNAs and putative target genes in seeds, and direct complementary association of them only in a case. Thus it is hard to say putative miRNA-targeted genes predicted here are surely involved in lipid metabolism and seed development, and I think the current title is based on many speculations.

  1. In luciferase activity assay (L212-223, L690-698), reporter and pre-miRNA constructs were co-transfected into what?

  1. It was hard for me to understand the explanation of 3p- and 5p-derived miRNAs (L156-160). I recommend to rephrase it to more reader-friendly sentences.

  1. Both figs. 5 and 6 explain GO enrichment on putative miRNA target genes. It seems redundant for me, and why don't you combine together?

Reviewer 2 Report

General comments

The authors have investigated small RNA dynamics in developing seeds of Camellia oleifera, and reported miRNAs involved in the regulation of several traits like seed size, lipid metabolism, growth and development etc. However, there are several concerns which should be addressed before the paper can be accepted.

The study lacks a very essential requirement for transcriptomic analysis. A minimum number of three biological replicates must be used for any transcriptome analysis. Unless the authors can provide any scientific reason as to why only two replicates were used for this study, data from one more replicate must be added. Also, the reproducibility of the data must be demonstrated using correlation analysis. Further, the grammar of the manuscript is very poor and several phrases are incorrect. Extensive revision of grammar must be undertaken. It is desirable that a standard English editing service is used.  Other than that, there are several fronts where the manuscript must be improved.

Specific comments

  1. In the abstract as well as the results, authors discuss negative and positive regulatory modules of miRNAs and their targets (pg 1 line number 23 to 25). This is unclear and misleading, as miRNAs regulate the target genes negatively. This must be rectified.
  2. Line number 46, ‘oleic acid accounts for over 75%’. Is it over 75% of oil content? Please clarify.
  3. Line number 62 – ‘till date’ please specify the month and year, as the readers may read the paper at different times.
  4. In the introduction, there are insufficient evidences discussed to explain the importance of miRNAs in regulating seed traits or lipid metabolism which is the main focus of this study. Instead, examples from leaf development and other traits are discussed. MiRNAs regulating seed development may be discussed here as examples.
  5. Line number 105 – Illumina technology analysis – analysis must be removed here.
  6. Line number 109 – what is meant by miRNA clusters? It is not clear.
  7. Line number 127 – relatively typical characters – please explain this phrase
  8. Line number 129 – what is meant by key periods?
  9. Line number 134 to 135 – Why only two replicates were used in this study?
  10. Line number 174 to 178 – the methodology of differential expression analysis is not explained here clearly. Which tool was used? What was the q value?
  11. Line number 180 – What were the threshold values used for the target identification?
  12. Line numbers 225 to 233 – It is not indicated how many sequences were obtained in the initial filtering steps after each step, like quality check, trimming, mapping to other RNAs etc. A flowchart may be made and shown in a figure in the main text or supplementary files.
  13. The classification of number of miRNAs according to number of reads is arbitrary (line number 257 to 262). Authors may refer to recent papers to classify them as low or highly expressed.
  14. The term coexpressed miRNAs has been incorrectly used. Expression of one miRNA in more than one samples does not mean that they are coexpressed. This needs to be checked throughout the manuscript.
  15. In the results, section 3.2, the characteristics of miRNAs reported must be elaborated. Also, the authors show in the figure 1 that the peak of miRNA lengths is at 24 nt. This is contrary to the facts known so far for miRNAs. The peak is normally expected at 21 nt for miRNAs while the peak at 24 nt is normally observed for siRNAs. Also, it is not clear whether this figure is actually made for miRNAs or the complete dataset. Please clarify the same in the legend and discuss the findings in the main text.
  16. Line number 287 – 295 The nomenclature of miRNAs is incorrect. Please refer to recent papers to correct it and devise the nomenclature according the the recent rules for your study.
  17. Figure 3 may better be replaced by a heat map instead of histogram to show the expression or fold changes.
  18. Table 1 may be shifted to the supplementary files as figure is sufficient to depict differentially expressed miRNAs
  19. Figure 4 needs to be remade without using a tool, it can be manually made in power point for better representation.
  20. Line number 333 conserved miRNA frequency is a misleading phrase. Also the use of terms like mdm, gma……..species is non-scientific.
  21. Line number 345 Which mRNA transcriptome data is used here? It is not clear.
  22. Line number 348 What is the q value used for GO and pathway enrichment? It must be mentioned.
  23. Line number 452 It is not clear why these target genes were considered crucial for regulation of seed size. Throughout the results, many genes have been mentioned to be important for trait regulation, but their function has not been stated properly and no citation has been given to mention the studies from where it was concluded. A table may be included in the supplementary describing the same and may be cited in the results section.
  24. Figure 8 Methodology of how heatmaps were made has not been discussed anywhere. Methods need to be  more elaborated.
  25. Why is the a part shown with a dendrogram but not the b part?
  26. Line number 526 How can there be a positive regulation effect of miRNA? This suggests that this gene is not actually targeted by miRNA as miRNAs always suppress the expression of their targets, even though it was predicted as a target.
  27. Figure 11 The labelling on the x axis showing the developmental stages is confusing, the stages should be clearly mentioned here.
  28. Figure 11 and 12 can be merged together by showing the relative expression of miRNAs and corresponding target genes in a single graph, as done in many recent papers.
  29. Discussion must be strengthened by explaining why particular candidates were prioritized.
  30. A conclusive figure must be included to finally enlist all the candidates/pathway categories and their roles in regulation of lipid metabolism and other seed traits.
  31. Extensive revision of grammar is needed. Some poor phrases are listed below:

Line number 13 has been being

Line number 16 in the critical periods of

Line number30 miRNAs and its target genes

Line number 51 tea oil camellia is unstable

Line number130 oil contents in these cultivars were in upward trend

And many others

These severely impact the readability and even the correctness of the manuscript and I therefore suggest usage of editing service.

Overall, all aspects of the manuscript including introduction, methods, results, discussion, conclusion, figures and tables need significant improvement.

Reviewer 3 Report

The manuscript “identification of miRNA-mRNA regulatory modules involved in lipid metabolism and seed development in a woody-oil tree (Camellia oleifera)” by Wu et al. is working on an important aspect of plant science by increasing knowledge on functional genomics in non-model plant species. Although being inherently more difficult to work with compared to a model plant species, it is the knowledge about those plant species which can provide a direct practical impact on e.g. agricultural applications. The authors aimed to investigate two cultivars with different seed oil content regarding their miRNA expression to find regulatory modes which might explain the difference in oil content seen in those cultivars.

The work can provide a solid basis for future studies, however, there are many issues which need to be addressed and due to the nature of experiments chosen the conclusions drawn from that study are inherently weak in nature. Let me summarize the critical points which led to my statement in the previous sentence.

Major points:

  1. The material and methods a lacking important information to fully understand and evaluate the scientific data presented.
    1. How was the mRNA transcript data generated or where does it come from? I assume it is publicly available data as no quality control of the sequencing results were described as well as library preparation etc. Please indicate the database the RNAseq data was retrieved from and how it was processed as well as analyzed.
    2. In line 166 the authors mention “with exclusion of specific species”. Which species where excluded? Please indicate and add a justification as to why this seems appropriate.
    3. The Dual-Luciferase assay method is missing a description into which cells of which species the reporter was transfected into. By using transfectant I assume it is either insect or mammalian cells.
  2. The identification of novel miRNAs has to follow certain essential criteria since many false positive hits can be derived by BLAST searching short reads and folding of flanking regions can form artefactual pseudo-hairpins which don’t exist in reality. Therefore, since Victor Ambros and colleagues first formulation of criteria for miRNA annotation in 2003, then the refinement to plants by Meyers et al. in 2008 and further updating to NGS-era by Axtell and Meyers in 2018, there are very defined criteria which need to be fulfilled to call a miRNA a novel miRNA. The authors present their own vague criteria in table S5, without any justification as to why they chose those criteria, however this still does not make the novel candidates trustworthy. There are many bioinformatic tools to more or less accurately predict novel miRNAs from small RNA sequencing data and the authors should use one of those programs. At least they confirmed expression signals for one such candidates, PC-3p-9080_426, via qRT-PCR, however this still does not validate the sequence as miRNA without providing a clear overview about all the necessary ancillary data needed.
    1. All those novel miRNAs have to be clearly labelled as potential miRNA-like sRNAs until further validation or meeting the above-mentioned criteria
    2. Why did the authors not chose to confirm the novel miRNAs in the dual-luciferase assay instead of miR393?
    3. Important to judge the validity of miRNA is the precision of the small RNA dicing product. If no prediction tool is used, at least a graphical representation of all mappable reads (with miRNA and miRNA* highlighted) on the predicted precursor should be presented in the supplementary data to judge the validity of the data.
  3. The presentation of figure 1 is wrong and needs to be exchanged. Small RNAs have distinct size classes measured in the length of nucleotides based on natural numbers. The use of a line diagram, especially one with an artificial fit, is misleading by creating peaks in decimal size ranges. It must be replaced by columns or any other scientifically correct representation. Also the labelling of the libraries should be consistent with the text (e.g. “M3” instead of “MY003”). Since miRNAs are mainly 21nt in size. Please indicate all nts on the x-axis or at least put minor ticks.
  4. The conclusions of the manuscript are based on the comparison of miRNA expression with mRNA expression of bioinformatically predicted targets. While this is valid to do for a descriptive overview, conclusions which drawn from this comparison are very limited at best! It is known that miRNAs and targets can have mutual exclusive expression patterns and might never encounter each other in vivo. Also targets can be regulated purely on the translational level which do not affect the mRNA transcript abundance. And even if negative correlation exists of a predicted pair, this is not a proof of actual miRNA cleavage action. The only way to confirm those interaction in the investigated system are degradome/RLM-RACE analysis or HITS-CLIP. While the latter is hardly doable with a non-model system and especially in plants the former is a common way to confirm miRNA/mRNA target regulation. Without those the authors must be much more careful in their interpretation of their data.
  5. In addition to point 4 the Dual-Luciferase assay does not add very much strength to the manuscript. It confirms the regulatory potential of the miRNA chosen but doesn’t mean this regulation happens within the studied plant at their given time-points without showing that those transcripts at least can partially co-exist within the same cells (e.g. by promoter-reporter lines or better feasible in non-model plants systems via in-situ hybridization).
    1. There is also a justification missing as to why the authors chose to investigate this particular miRNA-target pair via the luciferase assay.
    2. Why was the target pair identified by psRNA target while all the other predictions were done by Target Finder? Are the predicted miRNA bindings sites similar?

Minor comments

  1. The introduction is lacking information about the biogenesis of miRNAs (mainly DCL1) and their action (mainly via AGO1; translational inhibition and transcript cleavage) for a broader audience and to understand the implications for the study and be able to judge the presented data.
  2. Why choosing the Dual-Luciferase assay to be in non-plant cells? As assumed by use of transfectant. The processing of the precursor can be different between plants and insect/mammalian cells as plants have a processive DCL1 step using a single enzyme and mammals and insects have 2 enzyme Drosha/Dicer-system with distinct feature recognitions of the precursor.
  3. There is no explanation about the nomenclature of the discovered miRNAs with homology to known miRNAs. E.g. what does “1ss21GA” mean in ppe-miR172a-5p_1ss21GS.
  4. Tcc-miR162 has the exact same sequence as ath-miR162 which is a known auto-regulator of miRNA biogenesis by directly targeting DCL1. Did the miRNA target prediction came up with DCL-like targets? Could the differential expression of this miRNA between T1-T4 not be interesting for global regulation of miRNA biogenesis?
  5. For the qRT-data in Figure 11 and 12 please indicate the statistical test used for determining significance?